



# SNICAR-ADv4: a physically based radiative transfer model to represent the spectral albedo of glacier ice

Chloe A. Whicker[1], Mark G. Flanner[1], Cheng Dang[2], Charles S. Zender[3], Joseph M. Cook[4], and Alex S. Gardner[5]

[1]Department of Climate and Space Sciences and Engineering, University of Michigan, Ann Arbor, MI, USA
[2]Joint Center for Satellite Data Assimilation, University Corporation for Atmospheric Research, Boulder, CO, USA
[3]Department of Earth System Science, University of California, Irvine, CA, USA
[4]Department of Environmental Science, Aarhus University, Frederiksborgvej 339C, 4000, Roskilde, Denmark
[5]Jet Propulsion Laboratory, California Institute of Technology, Pasadena, CA 91109, USA

**Correspondence:** Chloe A. Whicker (cwhicker@umich.edu)

**Abstract.** Accurate modeling of cryospheric surface albedo is essential for our understanding of climate change as snow and ice surfaces regulate the global radiative budget and sea-level through their albedo and mass balance. Although significant progress has been made using physical principles to represent the dynamic albedo of snow, models of glacier ice albedo tend to be heavily parameterized and not explicitly connected with physical properties that govern albedo, such as the number and size of air bubbles, specific surface area (SSA), presence of abiotic and biotic light absorbing constituents (LACs), and characteristics of any overlying snow. Here, we introduce SNICAR-ADv4, an extension of the multi-layer two-stream delta-Eddington radiative transfer model with the adding–doubling solver that has been previously applied to represent snow and sea-ice spectral albedo. SNICAR-ADv4 treats spectrally resolved Fresnel reflectance and transmittance between overlying snow and higher-density glacier ice, scattering by air bubbles of varying sizes, and numerous types of LACs. SNICAR-ADv4 simulates a wide range of clean snow and ice broadband albedo (BBA), ranging from 0.88 for (30 μm) fine-grain snow to 0.03 for bare and bubble-free ice under direct light. Our results indicate that representing ice with a density of $650\,\mathrm{kg\,m^{-3}}$ as snow with no refractive Fresnel layer, as done previously, generally overestimates the BBA by an average of 0.058. However, because most naturally occurring ice surfaces are roughened "white ice", we recommend modeling a thin snow layer over bare ice simulations. We find optimal agreement with measurements by representing cryospheric media with densities less than $650\,\mathrm{kg\,m^{-3}}$ as snow and larger-density media as bubbly ice with a Fresnel layer. SNICAR-ADv4 also simulates the non-linear albedo impacts from LACs with changing ice SSA, with peak impact per unit mass of LACs near SSAs of $0.1\text{–}0.01\,\mathrm{m^2\,kg^{-1}}$. For bare, bubble-free ice, LACs actually increase the albedo. SNICAR-ADv4 represents smooth transitions between snow, firn, and ice surfaces and accurately reproduces measured spectral albedos of a variety of glacier surfaces. This work paves the way for adapting SNICAR-ADv4 to be used in land ice model components of Earth system models.

## 1 Introduction

Glaciers, ice caps, and ice sheets are large contributors to sea-level rise in our warming climate. These ice reservoirs, along with sea ice and seasonal snow, regulate the climate by altering the radiative budget through changes in surface albedo. Moreover, the local response of regional snow and ice to changing meteorology and climate is complicated and non-linear, making it increasingly important to model snow and ice surfaces using physical principles rather than empirically derived methods (Box et al., 2012; Budyko, 1969). In the last decade, ice sheets have become the dominant contributor to sea-level rise due to the increase in surface melt from the Greenland Ice Sheet (GrIS) (Bamber et al., 2018; Rignot et al., 2011; Goelzer et al., 2020; Hofer et al., 2020; van den Broeke et al., 2017). Such surface melt is governed by the

albedo of the ice sheet and local meteorology. The albedo
of snow and ice varies widely depending on the local atmo-
spheric conditions (Hofer et al., 2017), the light absorbing
constituents (LACs) present on the surface (Bøggild et al.,
2010; Skiles et al., 2018; Cook et al., 2020; Flanner et al.,
2007; Williamson et al., 2018; Tedstone et al., 2017; Marks
and King, 2014; Wang et al., 2004; Aoki et al., 2006), and the
metamorphic state of the snow and ice (Flanner and Zender,
2006; He and Flanner, 2020; Warren, 1982; Tedstone et al.,
2020; Aoki et al., 2000).

The albedo of the cryosphere varies with the spatial distri-
bution of snow, ice, and LACs and further evolves with the
melting of snowpack and glacier surfaces through the spring
and summer. As the snowline retreats during the melt season,
more bare ice is exposed. This bare ice has a lower albedo
and porosity than snow and therefore melts more and allows
for more runoff than snow, firn, and crustal surfaces (Ryan
et al., 2019; van den Broeke et al., 2017). This positive feed-
back has been referred to as the "snowline–albedo feedback"
and has been found to be the strongest seasonal melt am-
plifying feedback on the GrIS (Ryan et al., 2019). As polar
regions continue to warm, the length of the melt season is
expected to increase, exposing more ice (Jeffries et al., 2015;
Bøggild et al., 2010). As the snowpack melts and ice is ex-
posed, nutrients and liquid water become readily available,
allowing darkly pigmented glacier algae to colonize over the
ice surface. The annual blooms of glacier algae during the
spring and summer have been found to significantly reduce
the albedo of the GrIS and strongly contribute to surface melt
in the southwest ablation zone (Cook et al., 2020; Stibal et
al., 2017; Tedstone et al., 2020; Williamson et al., 2018; Yal-
lop et al., 2012). The spatial and temporal scale of these algal
blooms are expanding as higher summer temperatures result
in more bare ice exposure, available surface water, and nutri-
ents for glacier algae (Cook et al., 2020; Bøggild et al., 2010).
The ability to accurately model ice albedo and the effects of
glacier algae is critical for our understanding of future melt
and sea-level rise as (1) surface melt is modulated by albedo,
(2) the area of exposed bare ice is expanding under warmer
conditions, and (3) increased bare ice has the potential to fur-
ther reduce albedo through algal colonization.

The variations in snow albedo are well represented by
models. Snow is composed of small ice grains with high
albedo ranging from 0.7 to 0.9. The albedo of snow can be in-
fluenced by its physical properties, such as the grain size and
shape of the ice grains, the specific surface area (SSA), and
the thickness of the snowpack (Flanner and Zender, 2006; He
et al., 2017b; Saito et al., 2019; Dang et al., 2015). It can also
be influenced by environmental variables, such as the angu-
lar and spectral distribution of incident solar radiation, the
presence of clouds, and the presence of LACs (Gardner and
Sharp, 2010; Dang et al., 2015; Flanner and Zender, 2006).
Snow albedo has been accurately modeled using physical
principles to account for both its physical and environmen-
tal properties (Flanner and Zender, 2006; Dang et al., 2019;

Wiscombe and Warren, 1980; He and Flanner, 2020; War-
ren and Wiscombe, 1980; Gardner and Sharp, 2010). The
impacts of LACs, such as dust, black carbon, volcanic ash,
and pigmented snow algae, on snowpack albedo have also
been well studied and modeled (Flanner et al., 2007; War-
ren and Wiscombe, 1980; Skiles et al., 2018; Cook et al.,
2017; Painter et al., 2001; Flanner et al., 2014; Gardner and
Sharp, 2010; Flanner et al., 2021; Marks and King, 2014).
Bare glacial ice, on the other hand, which is frequently ex-
posed in glacial regions, is much darker than snow. Ice is
aged and compacted snow with an albedo ranging from 0.8
to 0.1, so it is more similar to a solid ice medium with air in-
clusions (Bøggild et al., 2010; Dadic et al., 2013; Mullen and
Warren, 1988; Briegleb and Light, 2007; Gardner and Sharp,
2010). The physical differences between snow and ice neces-
sitate distinct radiative transfer treatments, particularly with
regard to Fresnel reflectance and transmittance and scatter-
ing by air inclusions. However, multi-layer glacial bare ice
has not been modeled using physical principles to represent
the albedo of pure ice and the impact of Fresnel reflection
and transmission. Rather, ice albedo models are heavily pa-
rameterized using empirical data (Briegleb and Light, 2007;
van Kampenhout et al., 2020; van Dalum et al., 2020).

Various methods have been employed to model snow
albedo. These methods range from single layer two-stream
models to multi-stream statistical models. Snowpack is gen-
erally represented as a collection of independently scatter-
ing ice grains within an air medium. Snow radiative transfer
models (RTMs) utilize the optical properties of ice, the snow-
pack properties (density, thickness, and ice grain size and
shape), and the local atmospheric conditions to determine
the albedo of the entire snowpack (He and Flanner, 2020;
Flanner et al., 2007; Lee-Taylor and Madronich, 2002; Gard-
ner and Sharp, 2010; Libois et al., 2013; van Dalum et al.,
2019). Wiscombe and Warren (1980) developed a two-stream
delta-Eddington snow radiative transfer model for a ho-
mogenous snowpack. Flanner et al. (2007) utilized the two-
stream method developed by Toon et al. (1989) and devel-
oped the Snow, Ice, and Aerosol Radiative model (SNICAR),
a multi-layer heterogenous snow albedo radiative transfer
model that incorporates the influence of LACs on snow
albedo and is used in several Earth system models (ESMs),
such as the Community Earth System Model and the En-
ergy Exascale Earth System Model. Dang et al. (2019) found
that the Briegleb and Light (2007) delta-Eddington adding–
doubling radiative scheme calculates the albedo more ac-
curately than the Toon et al. (1989) solver. The Briegleb
and Light (2007) approach also allows for the inclusion
of refractive boundaries between snow–ice transitions. The
delta-Eddington adding–doubling solution iteratively calcu-
lates the reflectance and transmittance of each snow and ice
layer and the refractive boundary to then combine all lay-
ers to compute the total column optical properties (Briegleb,
1992; Briegleb and Light, 2007; Joseph et al., 1976; Coak-
ley et al., 1983). Dang et al. (2019) developed SNICAR-

AD by replacing the Toon et al. (1989) solving method with the delta-Eddington adding–doubling radiative method. Flanner et al. (2021) further developed SNICAR-AD (called SNICAR-ADv3) by including non-spherical snow grains, carbon dioxide snow, more types of LACs including snow algae, solar zenith angle (SZA)-dependent surface spectral irradiances, and extended spectral range (Flanner et al., 2021). However, these models are unable to represent glacier ice and heterogeneous snow and ice columns because they do not treat scattering by air bubbles, glacier algae, or Fresnel reflectance and transmittance across snow–ice or air–ice interfaces.

Representations of ice albedo, on the other hand, have historically been heavily parameterized to match empirical data. This simplification likely stems from the difficulty of representing an internal refractive boundary within a multilayer multiple scattering model and explicitly representing the optical properties of pure ice (Briegleb and Light, 2007; Mullen and Warren, 1988). Parameterizations of ice albedo range from spectrally constant approximations in ESMs to extending regional and offline snow RTMs using large snow grain sizes. For example, in the Community Earth System Model (CESM) ice albedo is 0.6 in the visible and 0.4 in the near-IR (van Kampenhout et al., 2020). Within the polar Regional Atmospheric Climate Model, van Dalum et al. (2020) parameterized the representation of bare ice on the GrIS by introducing impurities and increasing the ice grain size of the snow model to achieve satellite-observed albedo values for bare ice. In the offline SNICAR model, Cook et al. (2017) developed an option to use geometric optics, rather than Mie scattering (BioSNICAR_GO), to determine the optical properties of large snow grains and aspherical glacier algae, as ice grains are much larger than snow and glacier algae are more aspherical than snow algae (Cook et al., 2020). These parameterizations do not capture the low albedo of solid ice or variations in spectral albedo with changing ice conditions.

Some attempts have been made to explicitly simulate bare ice albedo. Mullen and Warren (1988) developed an offline model to find the optical properties of lake ice using information about the ice's microstructure, including the size distribution of air bubbles within the ice. They apply the delta-Eddington two-stream approximation to find the albedo of a single layer of lake ice topped with an interface that reflects and refracts light following Fresnel's laws. Dadic et al. (2013) expanded Mullen and Warren's model by adding a tuning parameter, which allowed their model to include the reflective interface when representing ice but remove it when representing snow. Gardner and Sharp (2010) utilized the 16-stream plane-parallel discrete ordinates radiative transfer (DISORT) model and applied it to a coupled snow, ice, and atmosphere system. They applied Mie theory to determine the optical properties of ice grains, air bubbles within ice, and light absorbing carbon. However, Gardner and Sharp's (2010) model does not account for Fresnel reflection from ice and is not readily applicable to ESMs

as it utilizes the 16-stream DISORT solver. The two-stream delta-Eddington multiple scattering parameterization developed by Briegleb and Light (2007) is the default sea-ice radiative transfer model within various ESMs. This model represents a multilayer heterogeneous snow and ice pack. Similar to the Mullen and Warren (1988) method, the Briegleb and Light (2007) model utilizes the delta-Eddington approximation but modifies it for any number of layers and incorporates an internal refractive interface. However, this model utilizes empirically derived inherent optical properties (IOPs) that are specific to sea ice and not applicable to glacier ice.

In this study, we combine and extend favorable elements of these previous RTMs to represent glacier ice. We explicitly represent scattering through the use of air bubbles, as in Mullen and Warren (1988) and Gardner and Sharp (2010), and we apply internal refraction across snow–ice interfaces, as in Briegleb and Light (2007), though with a more realistic spectrally resolved calculation. This model, the Snow, Ice, and Aerosol Radiative adding–doubling model version 4 (SNICAR-ADv4), simulates a heterogeneous multilayer snow and ice pack by explicitly resolving the microphysical optical properties of snow, ice, and LACs and performs radiative transfer calculations over the heterogenous cryospheric column. The following section describes the radiative transfer techniques applied in SNICAR-ADv4. Section 3 evaluates the model sensitivities and the impact of LACs and compares model outputs to in situ spectral albedo measurements.

## 2   Model description

SNICAR-ADv4 is a single-column heterogenous multilayer snow and ice model that explicitly represents the optical properties of snow, ice, and a range of biotic and abiotic light absorbing constituents. The model utilizes SNICAR's method for calculating the layer optical properties (Flanner et al., 2021). It treats snow as a collection of independently scattering ice grains within an air medium; thus, the bulk refractive index of a snow layer is equal to that of air. Ice is represented as independently scattering air bubbles within a solid ice medium with refraction that varies spectrally (Picard et al., 2016; Warren and Brandt, 2008). LACs are included as externally mixed and evenly distributed constituents in the snow and ice layers (Flanner et al., 2021). The model utilizes the radiative transfer equation in a plane-parallel media and applies the two-stream delta-Eddington solution to find the reflectance and transmittance of a single layer. While the plane-parallel approximation ignores horizontal variation and does not account for local slope and curvature of the surface, it is widely used in snow and ice RTMs (Flanner and Zender, 2005; Flanner et al., 2021; Dang et al., 2019; He and Flanner, 2020; Gardner and Sharp, 2010; Libois et al., 2013; Stamnes et al., 2000). Other studies have analyzed the sensitivity of albedo to sloped and rough surfaces and developed methods to account for surface roughness and slope (Picard

et al., 2020; Larue et al., 2020). SNICAR-ADv4 includes a Fresnel layer to account for the changing index of refraction between snow and ice layers. The Fresnel layer is a radiative layer with no thickness, which accounts for the bending of incoming solar radiation, the reflection of solar radiation at the refractive boundary, and the reflection and transmission of upwelling radiation beneath the refractive boundary (Briegleb and Light, 2007; Liou, 2002). The Fresnel layer is automatically placed directly above the first ice layer in a column. Once the reflectance and transmittance of each layer is known, the adding–doubling method is applied to combine each layer and find the total column radiative transfer solutions. These methods allow SNICAR-ADv4 to simulate a non-uniform multi-layer snow and ice column (as in Fig. 1).

## 2.1 Model parameters

SNICAR-ADv4 includes various tunable parameters for representing snow, ice, and LACs. The model includes three $H_2O$ ice refractive index datasets and the option to simulate $CO_2$ ice (Flanner et al., 2021). In this analysis, we utilize the Picard et al. (2016) and Warren and Brandt (2008) $H_2O$ ice refractive indices, as described in Flanner et al. (2021). The imaginary index of refraction as reported in Picard et al. (2016) is used from 0.2 to 0.6 µm, and the real and imaginary index of refraction reported by Warren and Brandt (2008) is used elsewhere in the spectrum. The model also simulates four different snow grain shapes: spheres, spheroids, hexagonal plates, and Koch snowflakes. It is recommended to use non-spherical grains because spheres produce unrealistically large scattering asymmetry parameters. This work defaults to the hexagonal plate shape as it has an intermediate asymmetry parameter between that of spheroids and Koch snowflake-shaped grains (Flanner et al., 2021; He et al., 2017b). SNICAR-ADv4 allows for the simulation of an arbitrarily thin snow layer overlying ice. This granular snow layer, or rough scattering layer, introduces surface roughness, as most naturally occurring ice surfaces have some degree of roughness from crustal surfaces or other small-scale irregularities (Briegleb and Light, 2007). A range of LACs are also included in SNICAR-ADv4. It includes all of the LACs that are present in SNICAR-ADv3 (four different dust species, volcanic ash, snow algae, and black and brown carbon) (Flanner et al., 2021) and adds darkly pigmented glacier algae found on the southwest GrIS (Cook et al., 2020). The user can specify the concentration of LACs and their vertical distribution within snow/ice layers. This allows for impurities to be concentrated on the uppermost layers of the column, which is typical of glacier snow and ice impurities, especially glacier algae colonies (Bøggild et al., 2010; Cook et al., 2020).

The model requires inputs regarding the environmental conditions: the solar zenith angle (SZA), downwelling spectral irradiance, and the spectral albedo of the underlying surface (e.g., bare ground albedo). It also requires information about the snow/ice column, including the snow grain or air bubble size distribution, the density of the snow or ice layer, and the thickness of each layer. All model inputs are outlined in Table 1. SNICAR-ADv4 requires all of the same inputs as SNICAR-ADv3 and adds inputs of the layer type (either snow or ice) and glacier algae properties (concentration, algae length, and algae width) (Cook et al., 2020). The new inputs specific to this version of SNICAR are indicated with an asterisk (*) in Table 1.

## 2.2 Radiative transfer solution

SNICAR-ADv4 begins by utilizing the optical properties of each individual constituent (snow grains, air bubbles, or LACs) within a modeled layer. The mass extinction cross sections ($\kappa_n$), asymmetry parameters ($g_n$), and single scattering albedos ($\omega_n$) for each constituent ($n$) are developed offline from Mie calculations using the Bohren and Huffman (1983) solving method (Flanner et al., 2021). The extinction optical depth ($\tau_n$) for each constituent is derived from the mass extinction cross section ($\kappa_n$) and the layer mass burden ($L_n$) (Flanner et al., 2021). For snow layers, $\omega_{snow}$, $\kappa_{snow}$, and $g_{snow}$ are derived using ice grain properties within an air medium. The Mie calculations for air bubbles within ice follow the same methodology except the relative refractive index of the scattering sphere is $N_{air}/N_{ice}$ instead of the inverse (Mullen and Warren, 1988; Gardner and Sharp, 2010; Dadic et al., 2013). Mullen and Warren (1988) hypothesize that Mie theory may not be generally applicable to scattering particles in an absorptive medium. However, in the visible part of the spectrum, ice is highly transparent so the absorbance has little influence, and for wavelengths >1.4 µm much of the light is absorbed before it is able to be scattered by an air bubble, so the scattering representation is less important at these wavelengths (Mullen and Warren, 1988).

The optical properties for ice layers are derived from the properties of air bubbles within an ice media and the ice absorptivity. The single scattering albedo ($\omega_{ice}$) and the mass extinction cross section ($\kappa_{ice}$) for ice are calculated differently than that of snow or LACs because they are specific to the volume of air within a layer. The layer bulk ice density is used to calculate the volume fraction of air ($V_{air}$) within each ice layer (Eq. 1), where $\rho_{blk}$ is the model input layer density, or bulk ice–air mixture density in kg m$^{-3}$, $\rho_{ice}$ is the density of pure ice (assumed here to be 917 kg m$^{-3}$); we neglect the mass of air in our calculations as in situ measurements of density do not include the mass of air.

$$V_{air} = \frac{\rho_{ice} - \rho_{blk}}{\rho_{ice}} \tag{1}$$

The spectrally varying mass absorption coefficient of ice ($\beta_a$, in units of m$^{-1}$) is found using Eq. (2), where $n_i$ is the spectrally varying ice imaginary refractive index (Picard et al.,

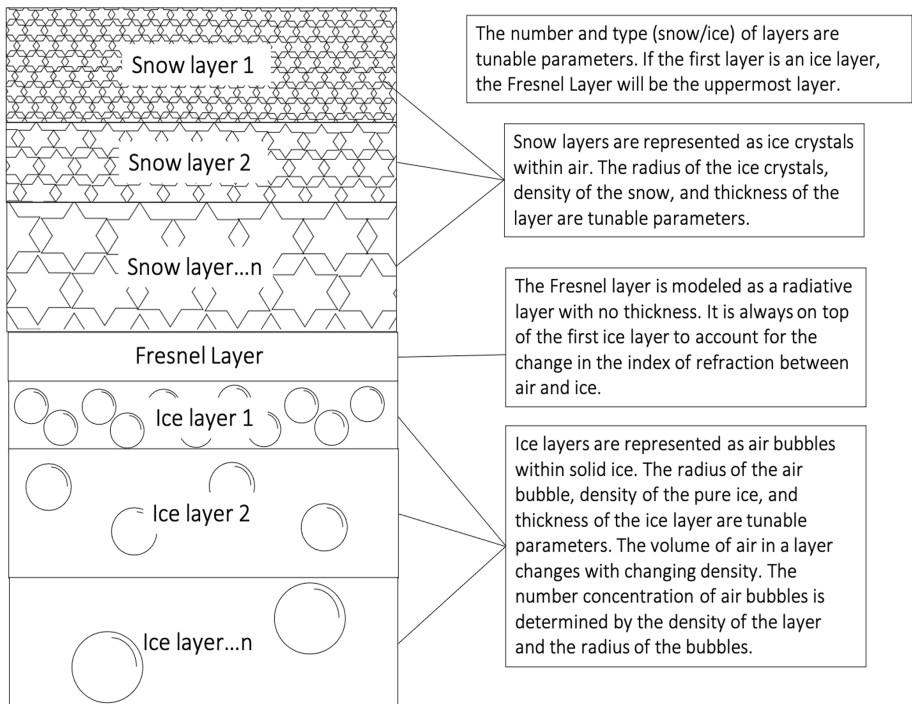

**Figure 1.** Model schematic of an example column of snow and ice.

2016; Warren and Brandt, 2008; Flanner et al., 2021).

$$\beta_{a,ice} = \frac{4\pi n_i}{\lambda} \tag{2}$$

The mass extinction cross sections ($\kappa_{ice}$, in units of m$^2$ kg$^{-1}$) and single scatter albedo ($\omega_{ice}$) are then derived from $\beta_a$ and $V_{air}$ following Eqs. (3) and (4) respectively.

$$\kappa_{ice} = \frac{\sigma_s V_{air}}{\rho_{blk}} + \frac{\beta_a}{\rho_{ice}}, \tag{3}$$

$$\omega_{ice} = \frac{\sigma_s V_{air}}{\rho_{blk}} \left(\frac{1}{\kappa_{ice}}\right), \tag{4}$$

where $\sigma_s$ is the volume scattering cross section (units of m$^2$ m$^{-3}$) determined using Mie calculations for a specific air bubble size distribution. The volume fraction of air, effective diameter ($d_{eff}$) of air bubbles, and the bulk density of the ice layer determine the specific surface area (SSA, $\alpha$, units of m$^2$ kg$^{-1}$), a measure of the total surface area of ice–air interfaces relative to the mass of ice:

$$\alpha = \frac{6V_{air}}{\rho_{blk}d_{eff}}. \tag{5}$$

If the size distribution of air bubbles is lognormal, as described in Carras and Macklin (1975) and qualitatively shown in Dadic et al. (2013), the total number concentration ($N_0$ units of m$^{-3}$) of air bubbles within each ice layer is related to the air volume fraction and effective bubble diameter as

$$N_0 = \frac{6V_{air}}{\pi d_{eff}^3} \exp(3\tilde{\sigma}_g^2), \tag{6}$$

where $\tilde{\sigma}_g$ is the geometric standard deviation of the lognormal distribution, assumed in this study to be $\ln(1.5)$. The value assumed for the lognormal width is not particularly important. This is because the optical properties of air bubble distributions with identical specific surface area (or effective radius) are nearly identical, and we use effective radius as the descriptive variable for bubble size. The distribution just needs to be sufficiently large enough to average over Mie resonance features. The specific surface area and bubble number concentration are both included as model outputs. Alternatively, the user can specify the $d_{eff}$ and SSA or $N_0$, from which $V_{air}$ is derived.

Once the optical properties of each constituent are calculated, the bulk layer properties ($\tau$, $\omega$, $g$) are derived following Flanner et al. (2021). After the bulk layer properties are calculated, they are delta scaled to account for the strong forward scattering by snow and ice (Briegleb and Light, 2007; Joseph et al., 1976). The Eddington two-stream solution then utilizes the delta-scaled bulk layer properties and the environmental model parameters (Table 1) to find the reflectivity and transmissivity of each layer (Shettle and Weinman, 1970). If the column contains an internal refractive boundary (between snow or air and the uppermost ice layer), the reflectivity and transmittivity of the refractive boundary are computed using the Fresnel formulas (Briegleb and Light,

**Table 1.** Description of each model input and output. The asterisk (*) indicates inputs that are unique to SNICAR-ADv4.

| Inputs | | Description |
|---|---|---|
| Environmental variables | Direct beam | Indicates use of direct beam or diffuse incident flux at the top of the column (values 0 or 1) |
| | Atmospheric profile | Loads atmospheric profiles from different locations on the Earth (values 1–7) |
| | Solar zenith angle | Cosine of the SZA |
| Snow and ice properties | Ice refractive index dataset | User can choose between three different refractive indices (Flanner et al., 2021) (values 1–3) |
| | Reflectance of the underlying surface | Spectrally varying or constant albedo of the surface underneath the column (values between 0 and 1, unitless) |
| | Layer thickness | The thickness of each layer in the column (units: m) |
| | Layer type* | Differentiates between snow and ice layers (snow layer is 1; ice layer is 2) |
| | Density | The density of each layer (cannot exceed 916.999) (units: $\mathrm{kg\,m^{-3}}$) |
| | Grain size | The radius of the snow grain or air bubble (units: μm) (values 10 to 20 000 TS1) |
| | Snow grain shape | Described in He et al. (2017b): 1 is sphere, 2 is spheroid, 3 is hexagonal plate, and 4 is Koch snowflake |
| | Snow shape factor | The ratio of non-spherical grain described in He et al. (2017b) |
| LAC properties | Mixing ratio of uncoated black carbon | units: $\mathrm{ng\,g^{-1}}$ |
| | Mixing ratio of coated black carbon | units: $\mathrm{ng\,g^{-1}}$ |
| | Mixing ratio of uncoated brown carbon | units: $\mathrm{ng\,g^{-1}}$ |
| | Mixing ratio of coated brown carbon | units: $\mathrm{ng\,g^{-1}}$ |
| | Mixing ratio of dust species | User can specify the size range and species of dust particles (units: $\mathrm{\mu g\,g^{-1}}$) |
| | Mixing ratio of ash species | User can specify the size range and species of ash particles (units: $\mathrm{\mu g\,g^{-1}}$) |
| | Snow algae concentration | Units: cells $\mathrm{mL^{-1}}$ |
| | Snow algae properties | User specifies the radius of the spherical algae cell and the concentration of pigments present (described in Flanner et al., 2021) |
| | Glacier algae concentration * | Units: $\mathrm{ng\,g^{-1}}$ |
| | Glacier algae properties * | User can specify the length and width of the aspherical algal cell (units: μm) |
| Outputs | | Description |
| | Spectral downwelling flux at the top of the column | Units: $\mathrm{W\,m^{-2}\,\mu m^{-1}}$ |
| | Broadband albedo | Weighted calculations for different spectral regions (solar, visible, near-IR) |
| | Spectral solar absorption | Calculated for each layer and different spectral regions (units: $\mathrm{W\,m^{-2}\,\mu m^{-1}}$) |
| | Spectral transmittance | Transmittance through the column (units: $\mathrm{W\,m^{-2}\,\mu m^{-1}}$) |

2007; Liou, 2002). The Briegleb and Light (2007) approach neglects the imaginary component of the refractive index in the Fresnel treatment and parameterizes the diffuse reflection by the layer to be spectrally constant. We alter the Briegleb and Light (2007) representation of the refractive boundary (Briegleb and Light, 2007, Eq. 22) to address those shortcomings.

We account for the effect of absorption on Fresnel properties by incorporating the complex refractive index of ice. We utilize the approach outlined by Liou (2002, Eq. 5.4.18). Liou (2002) applies the adjusted real and imaginary refrac-

tive indices to the Fresnel equations and Snell's law, where the spectrally varying adjusted real index of refraction ($N_r$) is

$$N_r = \frac{\sqrt{2}}{2}\left\{ m_{re}^2 - m_{im}^2 + \sin^2\theta_i + \left[ \left( m_{re}^2 - m_{im}^2 - \sin^2\theta_i \right)^2 + 4m_{re}^2 m_{im}^2 \right]^{1/2} \right\}^{1/2} \quad (7)$$

and $m_{re}$ and $m_{im}$ are the real and imaginary refractive indices of ice, respectively, and $\theta_i$ is the incident angle. The transmitted angle beneath the Fresnel layer in terms of the adjusted

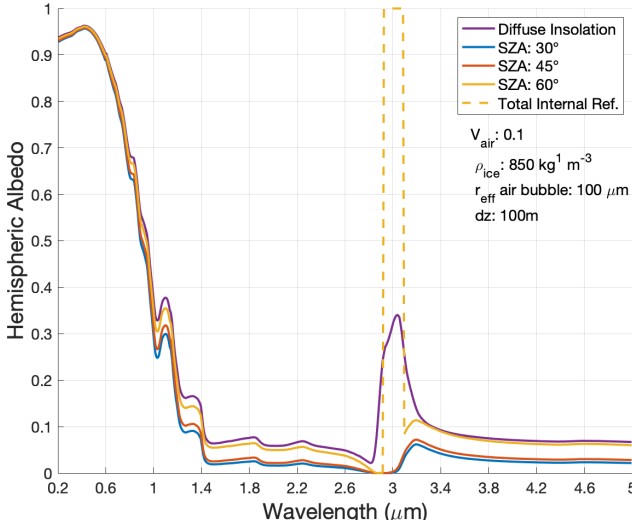

**Figure 2.** Spectral albedo with varying SZA. Each line indicates the spectral albedo simulated by SNICAR-ADv4 with a different SZA. The purple line shows the spectral albedo for the ice layer under diffuse light conditions, and the dashed line indicates where total internal reflection occurs on the smooth ice surface.

real refractive index is

$$\theta_t = \sin^{-1}\left(\frac{\sin\theta_i}{N_r}\right). \tag{8}$$

The Fresnel reflection and transmittance coefficients can be written in terms of the real (or adjusted real) index of refraction, where subscripts 1 and 2 indicate the perpendicular and parallel polarized components, respectively.

$$R_1 = \frac{\cos\theta_i - N_r\cos\theta_t}{\cos\theta_i + N_r\cos\theta_t} \quad T_1 = \frac{2\cos\theta_i}{\cos\theta_i + N_r\cos\theta_t} \tag{9}$$

$$R_2 = \frac{N_r\cos\theta_i - \cos\theta_t}{N_r\cos\theta_i + \cos\theta_t} \quad T_2 = \frac{2\cos\theta_i}{N_r\cos\theta_i + \cos\theta_t} \tag{10}$$

We include total internal reflection by the refractive boundary within SNICAR-ADv4 (Briegleb and Light, 2007). Total internal reflection influences radiation above the interface (in the air–ice transition) and radiation reflected back up on the boundary (ice–air transition) for SZAs greater than the critical angle of reflectance ($\theta_c$).

$$\theta_c = \sin^{-1}\left(\frac{m_{re,ice} - i\,m_{im,ice},}{m_{re,air} - i\,m_{im,air}}\right) \tag{11}$$

Total internal reflection occurs at wavelengths between 2.85 and 3.3 µm at SZA as low as 55° as seen in Fig. 2. It occurs for pure and smooth ice surfaces but is not realistic for naturally occurring ice which contains rough surfaces and impurities. We recommend imposing a rough scattering layer made up of snow grains to avoid total internal reflection.

We expand Briegleb and Light's (2007) diffuse reflection and transmission by the Fresnel layer to be spectrally varying. The spectrally varying diffuse reflection and transmission were developed offline following Briegleb and Light's (2007) method. We use Gaussian integration over a large number (10 000) of angles to integrate over the direct Fresnel reflection and transmission. We assume isotropic diffuse reflection and take into account the total internal reflection from above and below the Fresnel layer. The spectrally resolved diffuse reflection is stored offline to save computing time.

Once the reflectivity and transmittivity of each layer are found, SNICAR-ADv4 then combines each layer using the adding–doubling method and assuming any scattered radiation between layers is diffuse (Liou, 2002; Briegleb and Light, 2007). Finally, the spectral albedo and fluxes are computed from the total column reflectivity and transmittivity. SNICAR-ADv4's outputs include the spectral hemispheric albedo; the broadband albedo (BBA), which is the total albedo weighted by the incoming spectral irradiance; and the solar absorption of each layer and the entire column. A full list and description of each input and output variable are included in Table 1.

## 3 Model evaluation

In this section, we evaluate the model outputs and sensitivities by varying the snow, ice, and LAC properties and utilizing in situ spectral albedo measurements. First, we analyze the range of output albedos and the structure of the spectral albedos. Second, we analyze the influence of LACs. Lastly, we compare the model to snow and ice spectral albedo observations.

### 3.1 Model sensitivities

SNICAR-ADv4 simulates a wide range of spectral albedo that is consistent with measurements of snow and ice. Figure 3 shows the range in albedo due to changing effective snow grain radii (panel a) or air bubble radii (panels b–f), where higher albedo indicates a smaller snow grain or bubble size. Low-density media (less than 500 kg m$^{-3}$) are represented as snow, while high-density media (650 kg m$^{-3}$ and above) are treated as ice for the purpose of this comparison and sensitivity analysis. The albedo of both ice and snow reduces in the visible and near-IR parts of the spectrum as the radius of the snow grain or air bubble size increases, as smaller ice grains and bubbles scatter light more efficiently. The albedo declines as the density of ice increases and the volume of air decreases because air bubbles within ice are responsible for the scattering in the visible and near-IR parts of the spectrum. As the ice density increases, the influence of air bubble radius declines, as we can see in the reduction of shaded area with increasing ice density (Fig. 3) and the near-constant broadband albedo (BBA) for changing air bubble radius (Fig. 3g). As the radius of ice grains and air bubbles increases, the BBA declines, with less impact as the

grain/bubble size increases past $\sim 1000\,\mu m$ (Fig. 3b). Nearly pure ice (density of $916.999\,kg\,m^{-3}$ and volume fraction of $1.1 \times 10^{-6}$) has an almost constant spectral albedo around 0.03 (Fig. 3f and g). The reflection by very dense ice is due to the Fresnel reflection. The ice spectral albedo has a peak at $3.2\,\mu m$ due to the reflectivity of ice at normal incidence based on the spectrally varying indices of refraction utilized in SNICAR-ADv4. Besides this peak at $3.2\,\mu m$ we see minimal variability at wavelengths greater than $3\,\mu m$.

SNICAR-ADv4 simulates a wide range of BBA. Under the base case model input conditions (Tables 1 and A1) we find range in BBA from a high of 0.88 for small grain ($30\,\mu m$) snow to 0.03 for high-density ice ($916.999\,kg\,m^{-3}$) with sparse large air bubbles ($20\,000\,\mu m$) (Fig. 4). Figure 4 shows the BBA as a function of SSA, snow/ice density, and the number concentration of air bubbles. These ice properties are inherently interrelated as shown in Eqs. (1), (5), and (6) and can be described with the SSA of the snow/ice. The volume fraction of air can be expressed through the density, air bubble number concentration and size, and through the SSA. The albedo of ice varies with density (volume fraction of air), air bubble radius, and number concentration of air bubbles. To maintain a constant density with changing air bubble radius, the number concentration of air bubbles must change (Eqs. 1, 6). The BBA in Fig. 4 corresponds to the spectral albedo ranges in Fig. 3. Each spectrum within the shaded region of Fig. 3 corresponds to a single BBA and SSA value in Fig. 4. The blue $500\,kg\,m^3$ density line is represented as a single semi-infinite snow layer, and all other lines are represented as single semi-infinite ice layers, as indicated in Fig. 3. Figure 4a shows that as the SSA and BBA decrease, the radius of the snow grain/air bubble increases for a constant density. The leftmost (rightmost) end of each isopycnal line in Fig. 4a indicates a snow grain/air bubble radius of $20\,000\,\mu m$ ($30\,\mu m$). It is important to note that $20\,000\,\mu m$ is not a physically realistic snow grain or air bubble radius; this range of effective radii is utilized to analyze SNICAR-ADv4's total possible range in simulated albedo. Because SNICAR-ADv4 simulates wide ranges in albedo, it represents the transition between snow, firn, and ice SSAs smoothly (Fig. 4). Figure 4a also highlights that the SSA of snow and ice surfaces is a direct predictor of the BBA, which was also demonstrated by Gardner and Sharp (2010) and Dadic et al. (2013). Although Gardner and Sharp (2010) did not account for the Fresnel layer and Dadic et al. (2013) utilized a single layer column model, we still found a similar one-to-one relationship between BBA and SSA, and indeed it can be shown that in the geometric optics limit the mass scattering cross section of bubbly ice (first term of the right-hand side of Eq. 3) scales directly with SSA. Figure 4b shows the BBA achievable for constant densities by varying the ice grain/air bubble radius and the number concentration of air bubbles. Figure 4c demonstrates the range in BBA for constant air bubble radii. The black lines in Fig. 4c indicate a constant effective radius, and the colored dots show the corresponding density (and air fraction) for a given number concentration. For a constant bubble size, the BBA decreases with decreasing number concentration, indicating a reduction in the total volume of air. Note that the air bubble concentration is not included on the blue snow line in Fig. 4b and snow is not included in Fig. 4c, as there are no air bubbles in the snow representation.

The BBA of ice surfaces ($\rho \geq 600\,kg\,m^{-3}$) in Fig. 4 is lower than the BBA of snow surfaces with an equal SSA due to the incorporation of the refractive boundary between air and ice. The Fresnel layer between the air–ice media alters the modeled albedo, as it accounts for the light interaction with the refractive boundary as it moves into the ice medium (Fig. 5). SNICAR-ADv4 simulates a BBA difference of 0.0601 for ice layers with and without the Fresnel refractive boundary (for the particular model conditions outlined in Table A1, Fig. 5). When attempting to simulate the albedo of ice surfaces (with a density of $650\,kg\,m^{-3}$) using snow grains, rather than air bubbles within ice and a Fresnel layer, we found an average overestimation in BBA of 0.058. For snow and ice surfaces with a constant density and SSA, the difference in BBA ranges from 0.0368 for high-SSA ($\sim 40\,m^2\,kg^{-1}$) snow and ice to 0.0767 for low-SSA ($\sim 0.16\,m^2\,kg^{-1}$) snow and ice. The differences between snow and ice BBA (Fig. 4) and ice with and without a refractive boundary (Fig. 4) highlight the importance of accurately representing ice albedo. It also introduces different options for representing intermediate density firn. Other studies use large snow grains to produce the albedo of ice surfaces (Cook et al., 2017; van Dalum et al., 2020) or utilize the optical properties of air bubbles within ice but neglect the influence of the Fresnel layer (Gardner and Sharp, 2010). However, this work indicates that not accounting for the refraction between air and ice or treating ice as large grained snow may overestimate the broadband albedo of ice surfaces.

SNICAR-ADv4 also allows for the incorporation of a rough scattering layer, modeled as a thin snow layer, above ice that accounts for natural roughness of ice surfaces that contain a granular surface (Briegleb and Light, 2007). The thickness of the scattering layer and the snow grain size can be used to influence the modeled albedo and simulate columns that are similar to naturally occurring snow and ice conditions (Fig. 6). A scattering layer composed of fine ice grains ($100\,\mu m$), which is typical of freshly fallen snow, will generally increase the modeled albedo. With a scattering layer over a semi-infinite ice layer with a density of $850\,kg\,m^{-3}$ and a bubble radius of $100\,\mu m$ (Fig. 6a), we see the BBA increases with increasing scattering layer thickness, by up to 0.16 for a 10 cm thick layer. However, when large ice grains are used ($10\,000\,\mu m$), which are more similar to a coarse crustal surface found in nature, the scattering layer reduces albedo (Fig. 6c). In this case, the thickest rough scattering layer (10 cm) reduces the broadband albedo up to 0.07, as larger grain sizes reduce snow albedo (Flanner and Zender, 2006; Gardner and Sharp, 2010). It is important to note that the change in spectral albedo due to a thin scattering layer

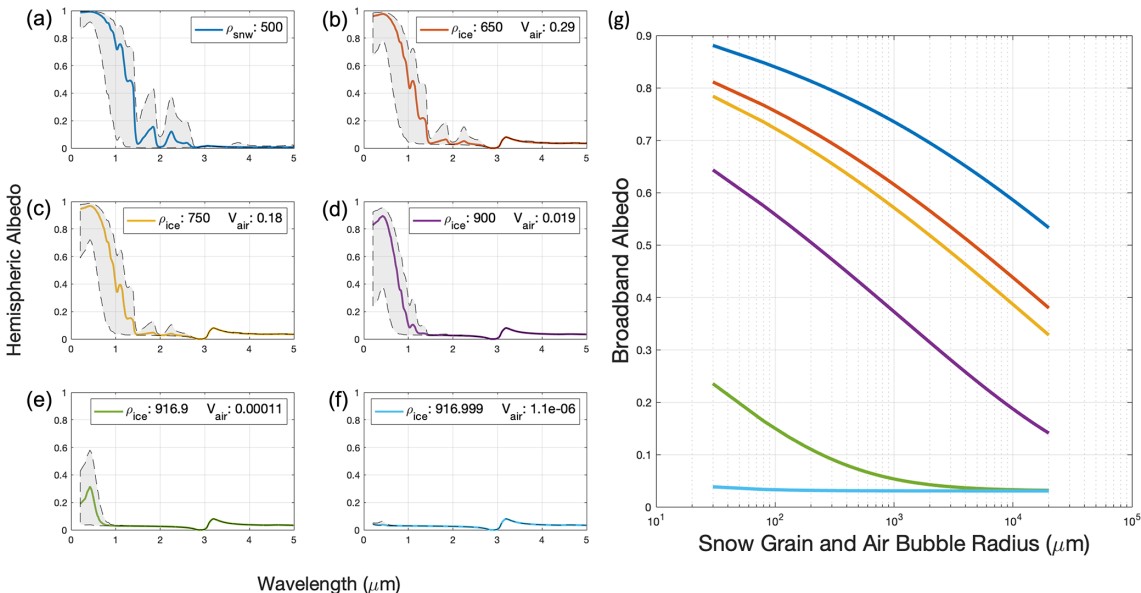

**Figure 3. (a–f)** Spectral albedo as a function of wavelength, snow or ice density, and the ice volume fraction of air. Shading indicates the full range of clean snow or ice albedo as a function of snow grain or air bubble radius, and the spectral albedo for an ice grain/air bubble with an effective radius of 180 μm is indicated by the colored line. Panel **(a)** is a snow layer; all the other panels are ice layers. The radius ranges from 30 μm (the highest albedo curves) to 20 000 μm (the lowest albedo curve). The model parameters are included in Table A1. **(g)** Broadband albedo (BBA) as a function of grain size (log scale). Each BBA and SSA corresponds to a spectral curve in panel **(a)** with a particular snow grain or air bubble radius.

depends on the ice layer conditions beneath it. The scattering layer can also be used to represent a "rotten" layer of white ice by including a thin layer of snow with a large grain size (Fig. 6c). Previous work has used large and aspherical snow grains to represent coarse ice in radiative transfer models (Cook et al., 2017; van Dalum et al., 2020). However, this work indicates that using snow grains to represent solid ice would generally overestimate the BBA.

We evaluate SNICAR-ADv4's ability to represent the spectrum of albedo produced by snow, firn, and ice surfaces with varying ice grain and air bubble sizes, as well as varying air concentrations. We found that SNICAR-ADv4 simulates a wide range of spectral albedo that is consistent with measurements of snow and ice. This analysis shows the spectral albedo of snow and ice as a function of its physical properties, such as density, volume of air in ice, and the radius of snow grains or air bubbles or the SSA. Firn has an intermediate density and can be treated as snow or ice, allowing for the techniques to be compared for media with equivalent SSAs. From a modeling perspective, it would be useful to specify a density threshold for representing a layer as snow or ice, as the model is sensitive to the ice density, and density is more easily measured in the field than other physical properties. Because ice is represented as air bubbles within snow it could be valid to treat all firn with a density greater than half that of pure ice ($458.5\,\mathrm{kg\,m^{-3}}$) as an ice layer. However, it is unlikely that ice that porous necessitates a refractive boundary. The transition between firn to ice is where pores between ice

grains close and form air bubbles within a solid ice media. The closing off of air bubbles occurs at an ice density around $\sim 830\,\mathrm{kg\,m^{-3}}$ or when $\sim 10\,\%$ of the ice volume is composed of air bubbles (Bender et al., 1997; Dadic et al., 2013). Because SNICAR-ADv4 incorporates numerous parameters, such as the density, grain size, layer depth, and the inclusion of a rough scattering layer, similar spectral albedos can be achieved using different model parameters (further described in Sect. 3.3.2). We see greater agreement between model and measurement for layers represented as ice with densities between $650$–$700\,\mathrm{kg\,m^{-3}}$ (Fig. 11b) and recommend that users treat media with densities over $650\,\mathrm{kg\,m^{-3}}$ as ice layers. To avoid unphysical results, we encourage users to implement a rough scattering layer above the Fresnel refractive boundary and to apply multi-layer schemes with gradually decreasing SSA with depth to ensure the snow/ice column is as realistic as possible (similar to the scheme outlined in Fig. 1).

## 3.2 Model LACs

This study also analyzes the impact of LACs on ice albedo. A full list of LACs included in SNICAR-ADv4 is included in Table 1, and they are further described by Flanner et al. (2021). The spectral influence of four types of externally mixed LACs (black carbon, dust, volcanic ash, and glacier algae) is presented in Fig. 7. Black carbon influences the albedo in the visible spectrum, with a strong absorption feature between 0.3 and 0.7 μm and a weak influence be-

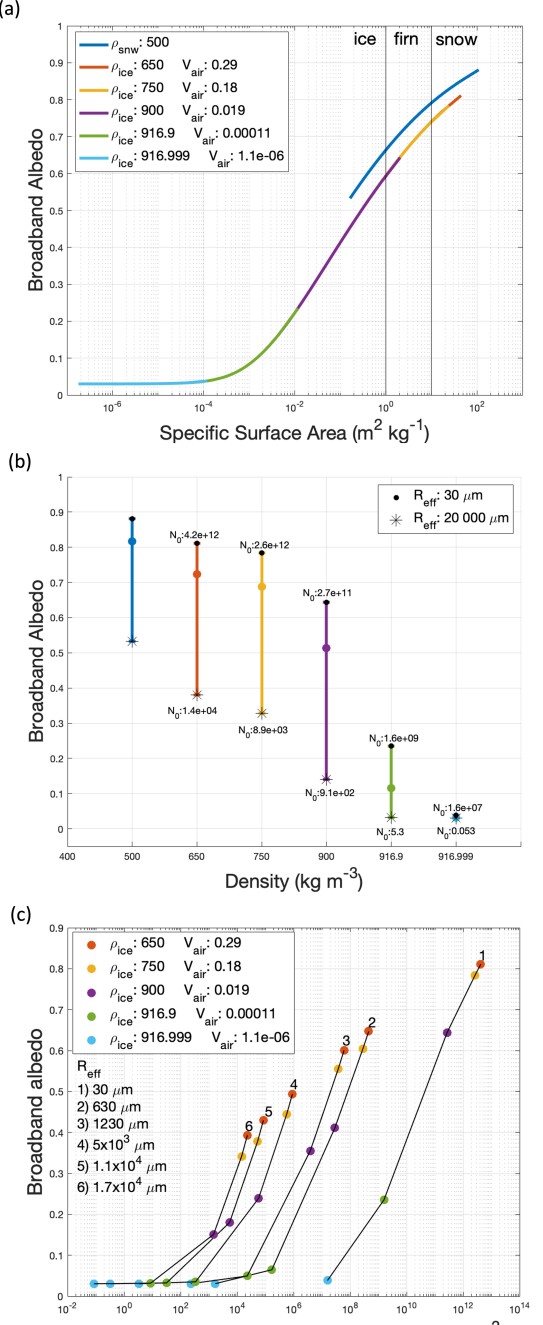

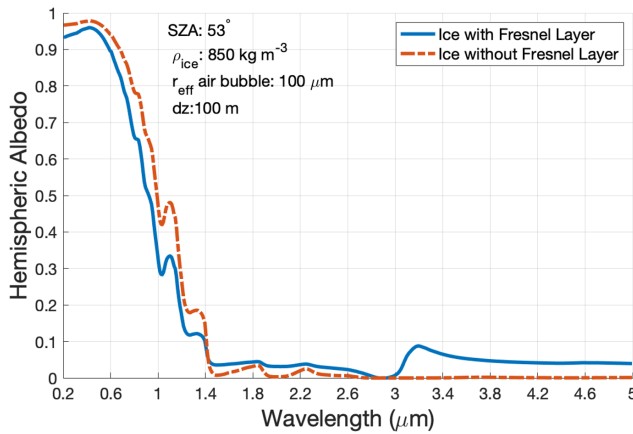

**Figure 5.** Ice spectral albedo with and without the refractive boundary between air and ice.

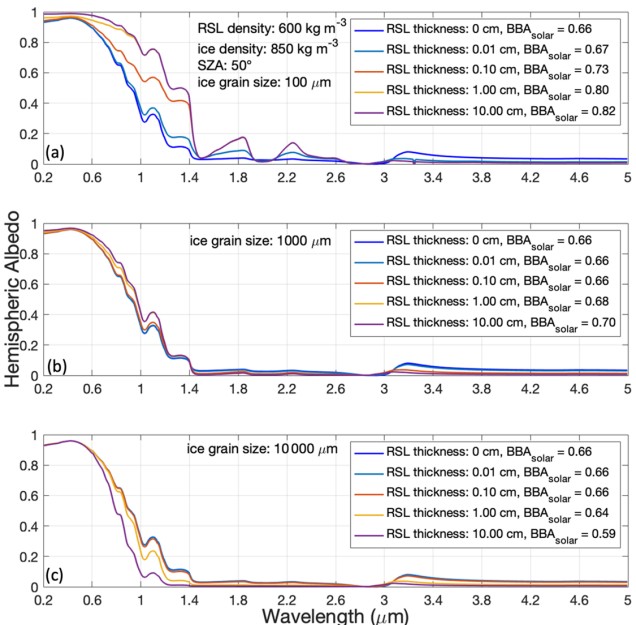

**Figure 6.** Varying rough scattering layer (RSL) thickness and snow grain radius. The model parameters are included in Table A1.

**Figure 4. (a)** Broadband albedo as a function of specific surface area, density, and volume fraction of air. The blue 500 density line is represented as snow; all other constant density lines are represented as ice. Model configuration is the same as Fig. 3 TS2. **(b)** Broadband albedo as a function of snow and ice density. Number concentration labels indicate the number concentration of air bubbles within the ice. Black stars indicate a bubble radius of 20 000 μm, black dots indicate a bubble radius of 30 μm, and colored dots indicate a bubble radius of 180 μm. **(c)** Broadband albedo as a function of number concentration of air bubbles. The lines indicate constant air bubble radius. The colored dots show where each corresponding ice density is reached for the combination of radius and number concentration.

tween wavelengths 1 and 1.4 μm (Fig. 7a). For nearly pure ice, black carbon actually increases the spectral albedo by ∼ 0.1 in the visible spectrum. GrIS dust and volcanic ash have similar effects on spectral albedo (Fig. 7b and c). GrIS dust absorbs more strongly than ash at wavelengths less than 0.5 μm and slightly less strongly between 0.5–0.7 μm. Dust causes a stronger albedo reduction and flattens the albedo curve at wavelengths less than 0.4 μm. Similar to black carbon, both dust and volcanic ash increase the albedo of dense dark ice in the visible wavelengths (Fig. 7b and c). Black carbon increases the visible albedo uniformly, while dust and ash increase the albedo most strongly around 0.7 μm (Fig. 7). Glacier algae reduces albedo most strongly in the

visible range of the spectrum, with the most unique absorptance spectra, due to the biotic pigments within the algal cell (Cook et al., 2020; Williamson et al., 2020). Glacier algae very weakly increases the albedo of the dark ice surface around 0.5 μm.

The impact of 100 ppb of black carbon on BBA varies with the density of the snow and ice and the snow grain and air bubble radius (Fig. 8). As the specific surface area of the snow and ice decreases, the same concentration of black carbon reduces the albedo more effectively, reaching a peak around a SSA of 0.1 $m^2 kg^{-1}$ in Fig. 8b. The peak in Fig. 8b indicates the maximum impact of 100 ppb of black carbon. Between SSAs of 0.01 and 0.001 $m^2 kg^{-1}$, we reach a plateau (Fig. 8a). In this SSA range, the impact of black carbon on BBA is nearly constant for varying ice density and air bubble radius. Once SNICAR-ADv4 reaches a SSA of $\sim 0.0007 m^2 kg^{-1}$, black carbon begins to increase the BBA of ice with densities greater than 916 $kg m^{-3}$. We see an increase of 0.042 in BBA due to scattering by black carbon within dark ice. GrIS dust and volcanic ash have similar effects on BBA (not shown). Similar to black carbon, we see a peak reduction in albedo and then a near-constant influence of these LACs until the LACs increase the BBA. When 100 ppm of GrIS dust is present, the peak reduction of BBA occurs at a SSA of 0.3 $m^2 kg^{-1}$ and increases the BBA for ice with a density of 916.999 $kg m^{-3}$ ($V_{air}$ of $1 \times 10^{-6}$ bubbles $m^{-3}$) by 0.047. Similarly, 100 ppm of volcanic ash reaches a peak influence around a SSA of 0.25 $m^2 kg^{-1}$ and increases the BBA for ice with a density of 916.999 $kg m^{-3}$ ($V_{air}$ of $1 \times 10^{-6}$) by 0.058. The peak in BBA change due to 1000 ppb of glacier algae (not shown) occurs around a SSA of 0.03 $m^2 kg^{-1}$ and only minorly increases the BBA by 0.0023 for a density of 916.999 $kg m^{-3}$ ($V_{air}$ of $1 \times 10^{-6}$) and an air bubble radius of 20 000 μm. It is important to note that high-density ice with large air bubble radii is not physically realistic, as a large fraction of the total air would be concentrated in few bubbles.

## 3.3 Model evaluation against measured spectral albedo

In this section, we compare SNICAR-ADv4 modeled albedo to four different sets of spectral albedo measurements of ice and high-density snow surfaces and quantify the difference between the model and measurement. The difference is the measurement value interpolated to the higher-resolution model wavelength scale, except in the case of the Cook et al. (2020) measurements which have a higher spectral resolution than the model. We subtract the measured value from the modeled albedo. Negative values indicate that the model is underestimating the albedo, and positive values indicate that the model is overestimating the albedo. The measurements were taken in Greenland, Antarctica, and Washington state. In some instances, snow and ice properties (such as density, grain size, and LAC concentrations) were also measured. If these variables were available, then they were implemented

within the SNICAR-ADv4 simulations. However, for most of the comparisons, the exact conditions are unknown. As a result, some of the model parameters are tuned to achieve good agreement between model and measurement. The SZA of each measurement is calculated based on the location and time of year the measurements were obtained using the NOAA Solar Calculator (https://gml.noaa.gov/grad/solcalc/, last access: 2 March 2022). We assumed samples were taken at solar noon to serve as a upper boundary for insolation. While the parameter choice might not be an exact representation of the physical and environmental conditions of each measurement, these results still demonstrate a range of realistic snow and ice albedos that can be recreated by SNICAR-ADv4. In addition, all model parameters used to achieve best fit are physically realistic and based on our understanding of snow and ice properties. For example, density and grain size increase with depth, and LAC decreases with depth. Where the physical properties of the snow and ice are unknown, the density and snow grain size are loosely based on normal ranges that have been measured in similar regions (Dadic et al., 2013; Cook et al., 2020; Carmagnola et al., 2013). All model parameterizations for the four comparisons can be found in Table A1; parameters that are well constrained by measurements have an asterisk.

### 3.3.1 Northeast Greenland Ice Sheet albedo measurements

Bøggild et al. (2010) obtained spectral albedo for a variety of snow and ice surfaces along the northeast ablation zone of the GrIS in Crown Prince Christian Land (Kronprins Christian Land). These measurements are particularly interesting because they span a wide range of spectral albedos, all observed in the northeast ablation zone, and highlight the importance of robustly simulating a wide range of snow and ice conditions. The spectral albedo was measured using a spectral radiometer. Bøggild et al. (2010) characterized the different snow and ice types and the LACs present on the surface. The model parameters used in these runs all had four layers of varying thickness, density, and snow or ice properties. The model parameters for the uppermost layer are indicated on each comparison (Fig. 9); all other model parameters are included in Table A1. Because Bøggild et al. (2010) did not include measurements of the snow and ice properties, such as density and snow grain size, for each albedo measurement, those model parameters were chosen to result in the most agreement between model and measurement. Bøggild et al. (2010) measured LACs that were present in the region. The LACs utilized in the model–measurement comparisons are constrained based on the impurities present in the measurements. Bøggild et al. (2010) reports locally sourced dust and small amounts of organic particulate matter. However, the exact concentration of the LACs is unknown. The LAC presence and size distribution included in the simulations are based on the Bøggild et al. (2010) qualitative descrip-

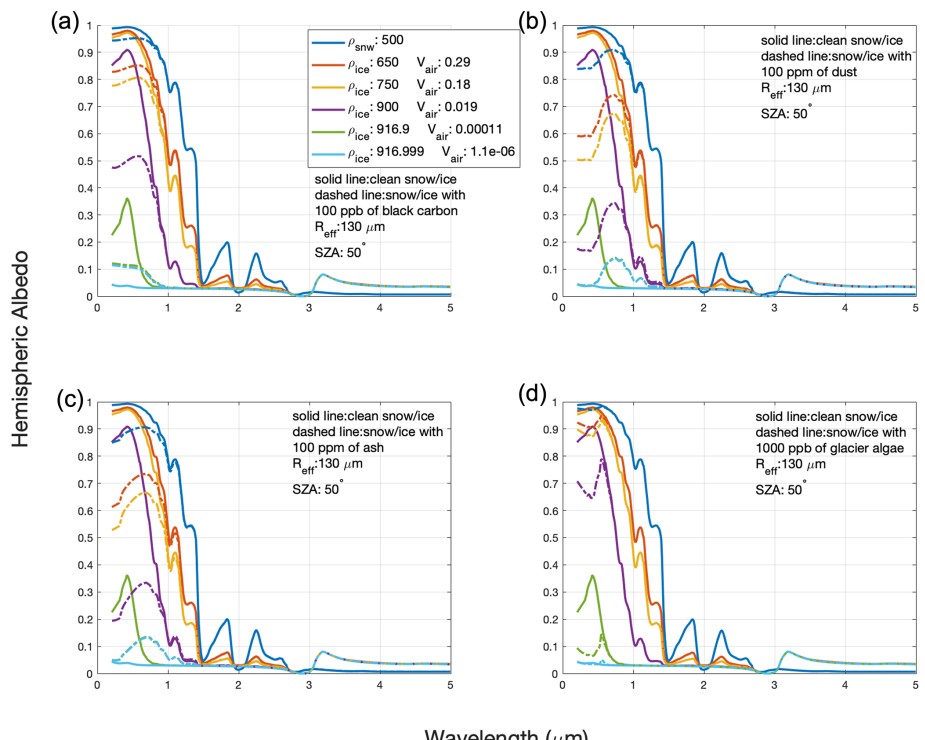

**Figure 7.** Model spectral albedo of various snow and ice surfaces with LACs. The model configuration of LAC-free surfaces (solid lines) is outlined in Fig. 3 TS3, and spectral curves are for a snow grain/air bubble radius of 130 μm. Dashed lines are the same with the addition of LACs. **(a)** Spectral albedo of various snow and ice media with and without 100 ppb of uncoated black carbon. **(b)** As in panel **(a)** but with and without 100 ppm of volcanic ash with a radius of 1.25–2.5 μm. **(c)** As in panel **(a)** but with and without 100 ppm of GrIS dust species with a radius of 1.25–2.5 μm. **(d)** As in panel **(a)** but with and without 1000 ppb of dry glacier algae with a length of 40 and width of 4 μm (Cook et al., 2020).

tions. The concentration of LACs, grain/air bubble size, and snow/ice density are tuned to find the best fit between model and measurement. We see good agreement for all six different ice types analyzed by Bøggild et al. (2010) (Fig. 9). In the visible, the maximum difference is 0.028. In the near-infrared (NIR), the difference ranges from −0.05 to 0.08. These results demonstrate SNICAR-ADv4's ability to reasonably simulate albedo using reasonable qualitative descriptions of the snow and ice surface.

### 3.3.2 Southwest Greenland Ice Sheet albedo measurements

Cook et al. (2020) took spectral albedo measurements and destructive biological samples in the southwest GrIS ablation zone in July 2017. Immediately after the spectral albedo measurements were taken, the surface of the snow or ice was removed and analyzed to quantify the concentrations of glacier algae and dust within the sample site. For the SNICAR-ADv4 comparison runs, we used the measured algal cell concentration and varied the snow and ice properties for a two-layer scheme. Cook et al. (2020) reported descriptions of the snow and ice surfaces and ice grain size for

two out of the four measurements used for comparison. We used the qualitative descriptions of the surface to determine if the top layer should be snow or ice, and we used an appropriate corresponding grain size where applicable. We see good agreement between the modeled and measured spectral albedo for samples with and without glacier algae, with maximal differences for snow in the NIR (Fig. 10). Glacier algae causes a large reduction in the visible part of the spectrum (Fig. 10d), which is replicated by SNICAR-ADv4 using the glacier optical properties developed by Cook et al. (2020). The largest discrepancies between SNICAR-ADv4 and the Cook et al. (2020) in situ measurements occur in regions with high algal concentrations (Fig. 10d). SNICAR-ADv4 underestimates the albedo in the visible for ice with high glacier algae concentrations; these discrepancies are likely due to uncertainty within the algae optical properties (Williamson et al., 2020; Cook et al., 2020).

### 3.3.3 East Antarctica albedo measurements

Dadic et al. (2013) measured spectral albedo, specific surface area, grain/bubble size, and the density of snow and ice in Allan Hills in East Antarctica. The optical measurements and

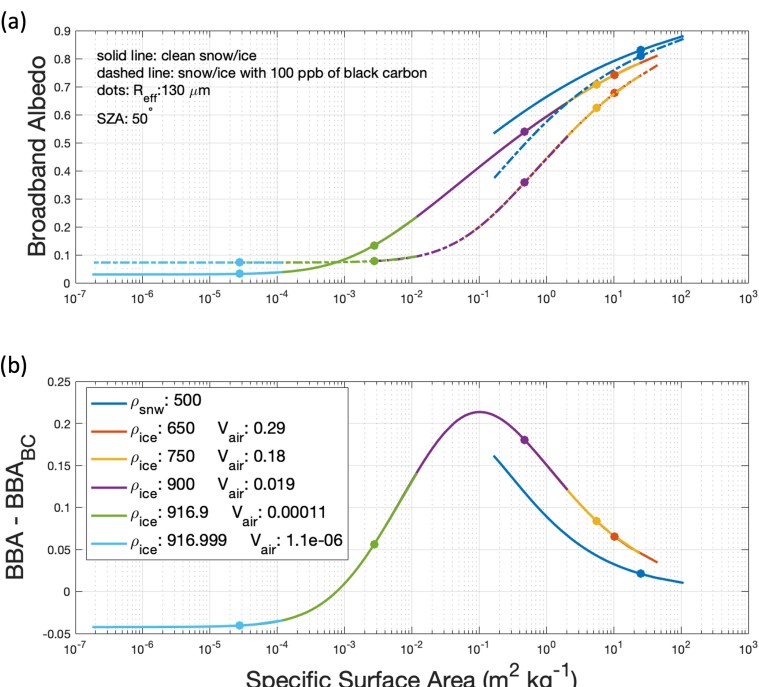

**Figure 8.** The impact of black carbon on snow and ice BBA as a function of SSA. Model configuration is the same as Fig. 7a. Dots indicate where the radius of the snow grain and air bubbles is 130 μm. **(a)** Broadband albedo as a function of specific surface area; solid lines indicate clean snow/ice and dashed lines indicate snow/ice with 100 ppb of black carbon. **(b)** The broadband albedo impact of black carbon shown as the difference between the sold and dashed lines in panel **(a)**.

recorded physical properties of the snow and ice make this dataset an extremely useful comparison. Dadic et al. (2013) developed an ice–air bubble model following the methodology of Mullen and Warren (1988) to compare to their measurements. Their model and measurements compare nicely. However, it achieves a smaller total range in albedo, it is only representative of a single layer, and does not include LACs. To compare SNICAR-ADv4 modeled spectral albedo to Dadic et al.'s (2013) measured spectral albedo, we utilized the measured SSA and density to find the effective ice grain and air bubble radii to constrain SNICAR-ADv4. The model configuration is two 10 m layers with properties measured by Dadic et al. (2013), but with the bottom layer having almost no influence on simulated albedo because the top layer is optically thick. For the clean snow and clean firn (Fig. 11a and b) the top layer is snow, and for white and blue ice (Fig. 11c and d) the top layer is ice with no scattering layer. In the case of clean snow, we see very good agreement in the visible part of the spectrum, with differences ranging from positive to negative 0.02. At wavelengths longer than 1 μm, we start to see more disagreement between the model and measurements. In the near-infrared (NIR) part of the spectrum, we see larger differences (ranging from −0.09 to 0.02) between the snow measurements and SNICAR-ADv4 (Fig. 11a). This is likely a result of the NIR albedo being highly sensitive to the snow grain size and shape in the top sub-millimeter of snow (Flanner et al., 2021). For the model comparison to firn

(Fig. 11b), we used both snow and ice layers to see which option achieved the best agreement. We found slightly better agreement when we represented both layers as ice with a visible improvement at wavelengths >1.2 μm. From around 1–1.8 μm we see about 0.04 less difference between the measurement and modeled ice than the modeled snow, indicating that the ice model implementation is better for higher-density media (650 kg m$^{-3}$). The model recreates the measured albedo quite accurately, with small variations occurring in the near-IR wavelengths for snow and visible for ice simulations (Fig. 11b–d). These results are particularly promising as they highlight SNICAR-ADv4's accuracy when empirical data on SSA, density, and bubble size are used to constrain the model.

### 3.3.4 Mount Olympus albedo measurements

Kaspari et al. (2015) measured spectral albedo and took ice cores to analyze the iron and black carbon concentration from snow and ice on Snow Dome on the Blue Glacier, Mount Olympus, in Washington state. In order to compare SNICAR-ADv4 to Kaspari et al.'s (2015) measured spectral albedo, we utilized their black carbon and iron measurements to include in the model as black carbon and dust. The other model parameters were estimated to find the best fit between the modeled and measured spectral albedo. The model configuration is three layers of snow and ice with varying densi-

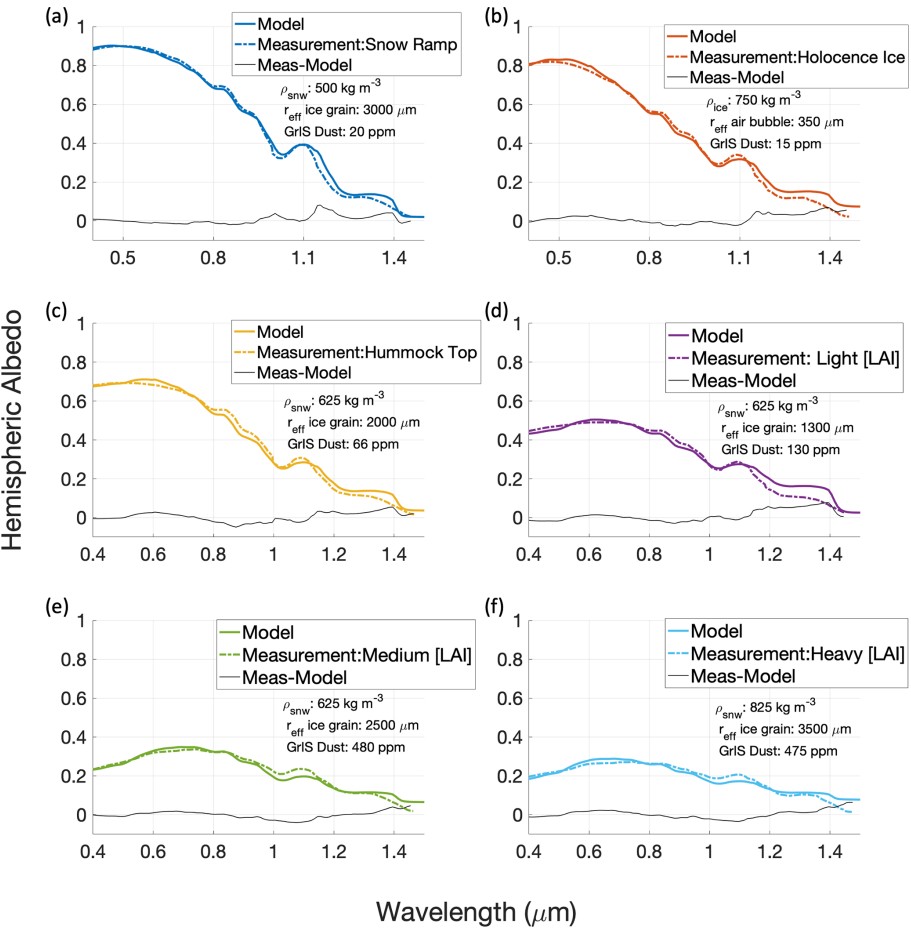

**Figure 9.** Spectral albedo measured by Bøggild et al. (2010) compared to SNICAR-ADv4 modeled spectral albedo. The model parameters are loosely constrained based on qualitative descriptions in Bøggild et al. (2010).

ties and thicknesses. In all four cases, the top layer is snow, and the bottom two layers are ice, except for the site 4 measurement comparison, where the middle layer is also snow. The top layer ranges from 1–2 cm in depth, the middle layer ranges from 5–50 cm, and the bottom layer is 5 m for all four comparisons. We see good agreement between modeled and measured spectral albedo for all four comparisons with slight deviations between 0.4–0.5 and 0.8–1.4 µm (Fig. 12). These deviations in the visible can likely be attributed to slight differences in the absorbance spectra of LACs, and in the NIR they are likely due to uncertainty in the grain size and shape.

## 4   Conclusions

Snow and ice surfaces regulate the global climate through changes in surface albedo. We have various advanced methods for simulating the dynamic albedo of snow surfaces using physical properties. Historically, however, ice albedo representations in models have been relatively simple and empirically based. SNICAR-ADv4 is a new snow and ice spectral radiative transfer model that utilizes the two-stream delta-Eddington adding–doubling solution. Key strengths of SNICAR-ADv4 are the broad range of physical properties it draws from to represent the albedo of snow and ice surfaces and the flexibility of the model to simulate non-uniform multi-layer cryospheric columns (example in Fig. 1). New features we have added to SNICAR-ADv4 that enable more realistic representation of glacier ice albedo include explicit representation of air bubbles, spectrally varying Fresnel reflection and transmittance at the ice–snow or ice–air interface, and glacier algae properties representative of those from southwest Greenland (Cook et al., 2020). The modeled albedo is dependent on user-specified layer properties, which include ice density, layer thickness, snow grain sizes and shapes (Flanner and Zender, 2006; He et al., 2017b); air bubble sizes and number concentrations (Dadic et al., 2013; Mullen and Warren, 1988; Gardner and Sharp, 2010); LAC concentrations (Cook et al., 2017; Polashenski et al., 2015; Skiles et al., 2017; Flanner et al., 2014; Wolff et al., 2009); and the environmental conditions, which include the surface spectral irradiance and SZA (Flanner et al., 2021). The new model simulates broadband albedos ranging from 0.03

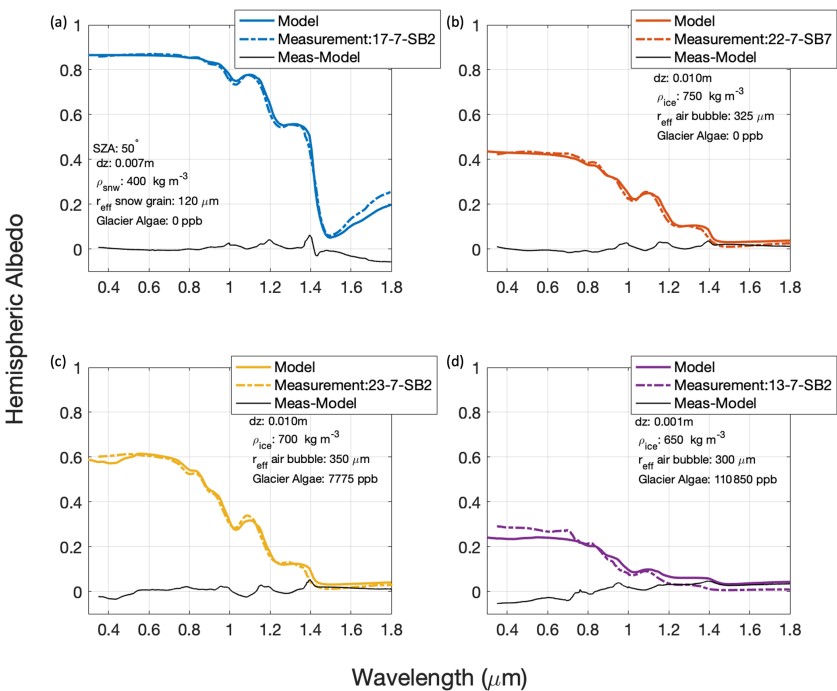

**Figure 10.** Spectral albedo measured by Cook et al. (2020) compared to SNICAR-ADv4 modeled spectral albedo. Model parameters are loosely based on qualitative snow and ice properties and quantitative algal cell measurements.

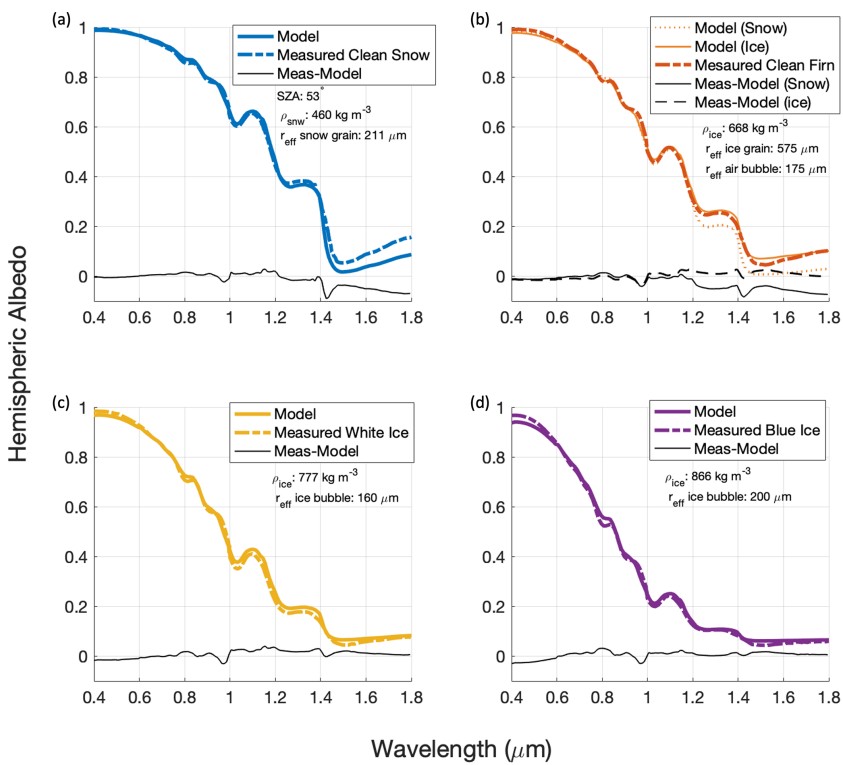

**Figure 11.** Spectral albedo measured by Dadic et al. (2013) compared to SNICAR-ADv4 modeled spectral albedo. The model parameters are tightly constrained with quantitative measurements made by Dadic et al. (2013).

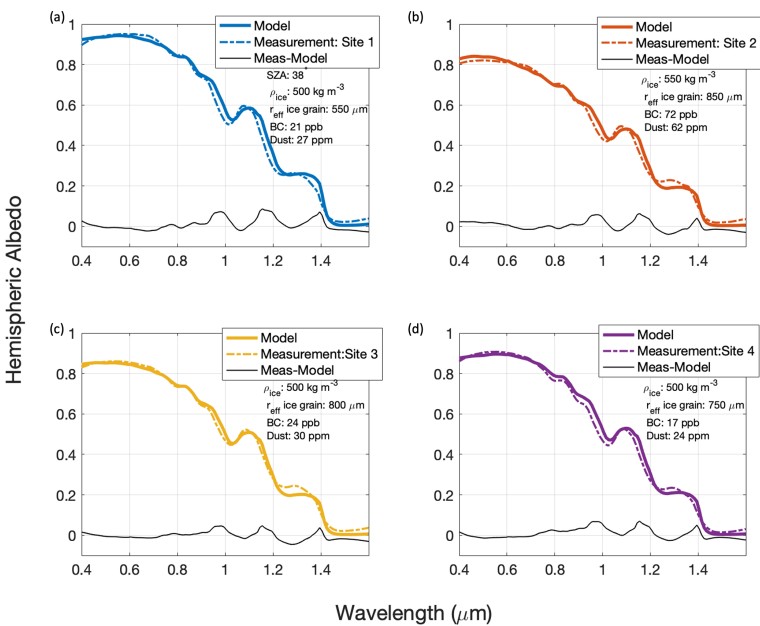

**Figure 12.** Spectral albedo measured by Kaspari et al. (2015) compared to SNICAR-ADv4 modeled spectral albedo. The model parameters for snow and ice properties are loosely based on qualitative descriptions, and the dust and black carbon concentrations are tightly constrained by quantitative measurements by Kaspari et al. (2015).

(bare, bubble-free ice) to 0.88 (fine-grained snow) (Figs. 3 and 4), with non-linear dependencies of LAC-induced albedo change on SSA. We compared model simulations to spectral measurements from four studies of widely varying ice conditions and LACs, finding good agreement in all cases, though ice physical properties were only well constrained in one of these studies (Dadic et al, 2013).

Future work needs to be done to analyze SNICAR-ADv4's performance over crustal or "rotten" ice surfaces, wet ice, and optically shallow ice. However, measurements of the spectral albedo and snow and ice properties for these surfaces are not readily available. SNICAR-ADv4 can also be extended to include weathered snow and ice as closely packed ice structures, as well as ponded water above the ice surface (Briegleb and Light, 2007; He et al., 2017a). LACs that are relevant over different regions can easily be added to SNICAR-ADv4 – for example, the optical properties of different dust or ash species that are common in other snow- or ice-covered regions. Future versions of SNICAR-ADv4 should also include the ability to simulate snow and ice surfaces with "clumped" spots of high LAC concentrations to more realistically simulate dark spots like cryoconite holes. It is increasingly important that we are able to simulate the full range of albedo for snow and ice surfaces in fully coupled global climate simulations to better quantify changes to the radiative budget and sea level. Because SNICAR-ADv4 is highly flexible, it is a promising new tool for improving our representation of the radiative properties of global snow and ice surfaces.

# Appendix A

**Table A1.** Model parameterizations for each run presented in this paper. NA indicates this parameter is not applicable for that particular run. TS4

| Figure | Model parameters |
|---|---|
| Base case<br>* all parameters are the same as the base case unless noted | incident flux: direct<br>SZA: 50°<br>d$z$: [100] m<br>grain size: [100] µm<br>layer type: [snow (1), ice (2)]<br>density: [500] kg m$^{-3}$<br>snow shape: hexagonal plate (3)<br>LACs: none<br>atmospheric profile: mid-latitude winter (1)<br>reflectance of the underlying surface: 0.25<br>ice refractive index: Picard et al. (2016) (3) |
| Figure 2 | incident flux: [diffuse, direct, direct, direct]<br>SZA: [NA, 30, 45, 60°]<br>layer type: [2]<br>density: [850] kg m$^{-3}$ |
| Figure 3a | grain size: [30–20 000] µm<br>layer type: [1]<br>density: [500] kg m$^{-3}$ |
| Figure 3b | grain size: [30–20 000] µm<br>layer type: [2]<br>density: [600] kg m$^{-3}$ |
| Figure 3c | grain size: [30–20 000] µm<br>layer type: [2]<br>density: [750] kg m$^{-3}$ |
| Figure 3d | grain size: [30–20 000] µm<br>layer type: [2]<br>density: [900] kg m$^{-3}$ |
| Figure 3e | grain size: [30–20 000] µm<br>layer type: [2]<br>density: [916.9 kg m$^{-3}$] |
| Figure 3f | grain size: [30–20 000] µm<br>layer type: [2]<br>density: [916.999] kg m$^{-3}$ |
| Figure 4 | same as Fig. 2 |
| Figure 5 | d$z$: [100] m<br>grain size: [100] µm<br>layer type: [2]<br>density: [850] kg m$^{-3}$ |
| Figure 6 | d$z$: [0–0.1, 100] m<br>grain size: [100–10 000, 100] µm<br>layer type: [1, 2]<br>density: [600, 850] kg m$^{-3}$ |
| Figure 7 | same as Fig. 3a–f TS5 except<br>100 ppb of uncoated black carbon<br>100 ppm of volcanic ash<br>100 ppm of GrIS dust species 3<br>1000 ppb of dry glacier algae |

https://doi.org/10.5194/tc-16-1-2022                                         The Cryosphere, 16, 1–24, 2022

Please note the remarks at the end of the manuscript.

| Figure | Model parameters |
| --- | --- |
| Figure 8 | same as Fig. 7a TS6 |
| Figure 9A (Bøggild et al., 2010, measurement/model comparison) | * SZA: 64°<br>d$z$: [0.005, 0.1, 0.5, 1] m<br>layer type: [1, 2, 2, 2]<br>density: [500, 625, 800, 850] kg m$^{-3}$<br>grain size: [3000, 800, 850, 900] μm<br>* GrIS dust size bin 2: [20, 0, 0, 0] ppm |
| Figure 9b (Bøggild et al., 2010, measurement/model comparison) | * SZA: 64°<br>d$z$ = [0.01, 0.5, 0.5, 1.0] m<br>layer type: [2, 2, 2, 2]<br>density: [750, 850, 850, 900] kg m$^{-3}$<br>grain size: [350, 425, 600, 600] μm<br>glacier algae: [200, 0, 0, 0] ppb<br>* GrIS dust size bin 2: [10, 0, 0, 0] ppm<br>* GrIS dust size bin 3: [5, 0, 0, 0] ppm |
| Figure 9c (Bøggild et al., 2010, measurement/model comparison) | * SZA: 64°<br>d$z$ = [0.01, 0.1, 0.5, 1.0 m]<br>layer type: [1, 2, 2, 2]<br>density: [750, 825, 850, 900] kg m$^{-3}$<br>grain size: [2000, 400, 600, 600] μm<br>* GrIS dust species 3 size bin 1: [10, 0, 0, 0] ppm<br>* GrIS dust species 3 size bin 2: [10, 0, 0, 0] ppm<br>* GrIS dust species 3 size bin 3: [10, 0, 0, 0] ppm<br>* GrIS dust species 3 size bin 4: [10, 8, 0, 0] ppm<br>* GrIS dust species 3 size bin 5: [10, 8, 0, 0] ppm |
| Figure 9d (Bøggild et al., 2010, measurement/model comparison) | * SZA: 64°<br>d$z$: [0.001, 0.1, 0.5, 1.0] m<br>layer type: [1, 2, 2, 2]<br>density: [650, 860, 905, 915] kg m$^{-3}$<br>grain size: [31 300 TS7, 650, 750, 800] μm<br>* GrIS dust species 3 size bin 1: [5, 0, 0, 0] ppm<br>* GrIS dust species 3 size bin 2: [5, 0, 0, 0] ppm<br>* GrIS dust species 3 size bin 3: [15, 10, 0, 0] ppm<br>* GrIS dust species 3 size bin 4: [15, 15, 0, 0] ppm<br>* GrIS dust species 3 size bin 5: [35, 30, 0, 0] ppm |
| Figure 9e (Bøggild et al., 2010, measurement/model comparison) | * SZA: 64°<br>d$z$: [0.0005, 0.1, 0.5, 1.0] m<br>layer type: [1, 2, 2, 2]<br>density: [750, 875, 900, 910] kg m$^{-3}$<br>grain size: [2500, 600, 700, 800] μm<br>* GrIS dust species 3 size bin 1: [50, 0, 0, 0] ppm<br>* GrIS dust species 3 size bin 2: [50, 0, 0, 0] ppm<br>* GrIS dust species 3 size bin 3: [50, 30, 0, 0] ppm<br>* GrIS dust species 3 size bin 4: [100, 50, 0, 0] ppm<br>* GrIS dust species 3 size bin 5: [100, 50, 0, 0] ppm |

| Figure | Model parameters |
| --- | --- |
| Figure 9f (Bøggild et al., 2010, TS8 measurement/model comparison) | * SZA: 64°<br>d$z$: [0.0005, 0.1, 0.5, 1.0] m<br>layer type: [1, 2, 2, 2]<br>density: [825, 900, 910, 915] kg m$^{-3}$<br>grain size: [3500, 750, 850, 950] μm<br>* GrIS dust species 3 size bin 1: [75, 0, 0, 0] ppm<br>* GrIS dust species 3 size bin 2: [75, 0, 0, 0] ppm<br>* GrIS dust species 3 size bin 3: [75, 25, 0, 0] ppm<br>* GrIS dust species 3 size bin 4: [75, 50, 0, 0] ppm<br>* GrIS dust species 3 size bin 5: [75, 25, 0, 0] ppm |
| Figure 10a (Cook et al., 2020, measurement/model comparison) | * SZA: 59°<br>d$z$: [0.007, 0.01] m<br>* layer type: [1, 2]<br>density: [400, 750] kg m$^{-3}$<br>* grain size: [120, 600] μm<br>* glacier algae: [0, 0] ppb |
| Figure 10b (Cook et al., 2020, measurement/model comparison) | * SZA: 59°<br>d$z$: [0.01, 0.08] m<br>* layer type: [2, 2]<br>density: [750, 900] kg m$^{-3}$<br>grain size: [325, 700] μm<br>* glacier algae: [0, 0] ppb |
| Figure 10c (Cook et al., 2020, measurement/model comparison) | * SZA: 59°<br>d$z$: [0.01, 0.05] m<br>* layer type: [2, 2]<br>density: [700, 750] kg m$^{-3}$<br>* grain size: [350, 800] μm<br>* glacier algae: [7775, 0] ppb |
| Figure 10d (Cook et al., 2020, measurement/model comparison) | * SZA: 59°<br>d$z$: [0.0005, 0.05] m<br>* layer type: [2, 2]<br>density: [650, 875] kg m$^{-3}$<br>grain size: [300, 900] μm<br>* glacier algae: [110 850, 0] ppb |
| Figure 11a (Dadic et al., 2013, measurement/model comparison) | * SZA: 53.25°<br>d$z$: [10, 10] m<br>* layer type: [1, 2]<br>* density: [460, 894] kg m$^{-3}$<br>* grain size: [211, 525] μm |
| Figure 11b (Dadic et al., 2013, measurement/model comparison) | * SZA: 53.25°<br>d$z$: [10, 10] m<br>layer type: snow [1, 2], ice [2, 2]<br>* density: [668, 894] kg m$^{-3}$<br>* grain size: snow [575, 525], ice [175, 525] μm |

| Figure | Model parameters |
|---|---|
| Figure 11c (Dadic et al., 2013, measurement/model comparison) | * SZA: 53.25° <br> d$z$: [10, 10] m <br> layer type: [2, 2] <br> * density: [777, 894] kg m$^{-3}$ <br> * grain size: [160, 525] μm |
| Figure 11d (Dadic et al., 2013, measurement/model comparison) | * SZA: 53.25° <br> d$z$: [10, 10] m <br> layer type: [2, 2] <br> * density: [866, 894] kg m$^{-3}$ <br> * grain size: [200, 525] μm |
| Figure 12a (Kaspari et al., 2015, measurement/model comparison) | * SZA: 37.5° <br> d$z$: [0.015, 0.5, 5] m <br> layer type: [1, 2, 2] <br> density: [500, 500, 900] kg m$^{-3}$ <br> grain size: [550, 100, 400] μm <br> * Sahara dust species 4: [27, 0, 0] ppm <br> * uncoated black carbon: [21, 0, 0] ppb |
| Figure 12b (Kaspari et al., 2015, measurement/model comparison) | * SZA: 37.5° <br> d$z$: [0.01, 0.05, 5] m <br> layer type: [1, 2, 2] <br> density: [550, 675, 850] kg m$^{-3}$ <br> grain size: [850, 650, 700] μm <br> * Sahara dust species 4: [62, 0, 0] ppm <br> * uncoated black carbon: [72, 0, 0] ppb |
| Figure 12c (Kaspari et al., 2015, measurement/model comparison) | * SZA: 37.5° <br> d$z$: [0.015, 0.1, 5] m <br> layer type: [1, 2, 2] <br> density: [500, 700, 910] kg m$^{-3}$ <br> grain size: [800, 500, 550] μm <br> * Sahara dust species 4: [30, 0, 0] ppm <br> * uncoated black carbon: [24, 0, 0] ppb |
| Figure 12d (Kaspari et al., 2015, measurement/model comparison) | * SZA: 37.5° <br> d$z$: [0.02, 0.1, 5] m <br> layer type: [1, 1, 2] <br> density: [500, 650, 910] kg m$^{-3}$ <br> grain size: [750, 1500, 550] μm <br> * Sahara dust species 4: [24, 0, 0] ppm <br> * uncoated black carbon: [17, 0, 0] ppb |

*Code availability.* The code used in this paper is available on GitHub at https://github.com/chloewhicker/SNICAR-ADv4 (last access: 26 August 2021). The exact version of the code and input files used to create the figures and results presented here can be accessed at https://doi.org/10.5281/zenodo.5270383 (Whicker, 2021). CE1

*Data availability.* All model data can be simulated using the SNICAR-ADv4 code and the model parameters in Table A1. The spectral albedo measurements in Figs. 9–12 can be found in their corresponding citations as follows: CE2

- Fig. 9 – Bøggild et al. (2010) Fig. 5;

- Fig. 10 – Cook et al. (2020), https://doi.org/10.5281/zenodo. 3564501 (Cook, 2019) TS9;

- Fig. 11 – Dadic et al. (2013), https://digital.lib.washington. edu/researchworks/handle/1773/37324 TS10 (last access: 16 March 2022);

– Fig. 12 – Kaspari et al. (2015) Figs. 2–3.

*Author contributions.* CAW wrote the manuscript, merged and edited relevant code, performed sensitivity analyses, and compared modeled albedo to spectral albedo measurements. MGF designed the study; provided Mie optical properties for snow, ice, and all LACs except glacier algae; and oversaw the research. JMC provided the glacier algae optical properties utilized in the model. CD and CSZ incorporated the adding–doubling solver into SNICAR. ASG contributed to the model development process for representing ice as air bubbles within an ice medium. All authors reviewed the manuscript.

*Competing interests.* The contact author has declared that neither they nor their co-authors have any competing interests.

*Acknowledgements.* We thank Ruzica Dadic, Carl Bøggild, Susan Kaspari, and Joseph Cook for their snow and ice spectral albedo measurements.

*Financial support.* This work was supported by grants OPP-1712695 and DGE 1841052 from the U.S. National Science Foundation. Charles S. Zender and Cheng Dang were supported by the Energy Exascale Earth System Model (E3SM, formerly ACME) project, funded by the U.S. Department of Energy, Office of Science, Office of Biological and Environmental Research (grant nos. DE-SC0012998 and LLNL-B639667) TS11 . Joseph M. Cook acknowledges ERC Synergy Grant (856416) Deep Purple.

*Review statement.* This paper was edited by Thomas Mölg and reviewed by Ruzica Dadic and one anonymous referee.

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

**Remarks from the language copy-editor**

CE1    Please verify and confirm the changes to this section.
CE2    Please verify and confirm the changes to this section.

**Remarks from the typesetter**

TS1    Thank you for your feedback. Please note that it is our standard to display five figure numbers as it is. If this is a decimal, a full stop should be added rather than a comma. Please check.
TS2    Please confirm.
TS3    Please confirm.
TS4    Please confirm adjustments in the table.
TS5    Please confirm.
TS6    Please confirm.
TS7    Please give an explanation of why this needs to be changed. We have to ask the handling editor for approval. Thanks.
TS8    Please confirm.
TS9    Please confirm DOI and citation.
TS10   Please check the link and provide reference list entry including creators, title, repository, and date of last access.
TS11   Please confirm adjustment.
TS12   Thank you for your feedback. Please note that the addition "[data set]" or "[code]" are only reserved for data repositories.
TS13   Please confirm addition of "ISBN".
TS14   Please confirm addition.
TS15   Please confirm reference list entry.
TS16   Please confirm article number.
TS17   Please check if this is correct.
TS18   Please confirm.
TS19   Please confirm reference list entry.
TS20   Please note: D13 has not been added since this is the issue number which is not needed.
TS21   Please confirm name and initials.
TS22   Please confirm article number (see https://agupubs.onlinelibrary.wiley.com/action/showCitFormats?doi=10.1029/2009JE003350).