# Peer review of "SNICAR-ADv4: A physically based radiative transfer model to represent the spectral albedo of glacier ice"

_The Cryosphere, 2021_

## Author Response (AR1)

*Thank you for the helpful and thorough comments. All comments are addressed below, with the original comment and the response in italics.*

Review #1 (Anonymous):

Summary and recommendations

This manuscript describes a physically based radiative transfer model (RTM) to represent the spectral albedo of glacier ice, called SNICAR-ADv4. In this RTM, air bubbles in the ice were treated as a scatterer, the layer structure was considered, and the ice and the snow overlying the ice were coupled, resulting in that the RTM of coupled atmosphere-snow-ice system was completed as SNICAR-ADv4. It can be said that we have taken a step forward to understand the spectral albedo of glacier surface. This is a very important research for the climate studies using SNICAR-ADv4. Reviewer gives a certain appreciation for reasons mentioned above. However, there are some concerns regarding the explanation and technical details of the validation method of SNICAR-ADv4, which are list below. Given this, I recommend this paper for publication after minor revisions with attention to comments.

Major comments

1. Regarding the surface roughness, called "the surface scatter layer (SSL)" which consists of "Fresnel layer" plus "thin snow layer". Reviewer guesses that the SSL is to avoid a specular reflection in the flux calculations. This is a good representation but the SSL is the approximate one even in a physical-based method. There are some more realistic models proposed (e.g Stamnes et al., 2011, Sun et al., 2019). In order to confirm this representation properly, reviewer recommends to depict the solar zenith angle dependence on the spectral albedo and then explain it at high solar zenith angle.

*You are correct in that the SSL is used to avoid specular reflection over ice surfaces. We see total internal reflection around 3 μm, where the refractive index of ice drops below that of air, for SZA greater than ~55 °. As requested, we've added a figure showing spectral albedo at different SZA (Figure 1 in supplement), and the revised manuscript will include the figure and a description of influence of the Fresnel layer at λ ~ 3 μm at high SZA.*

[Figure]

2. Authors described the effects of the air bubble and the SSL on the spectral albedo especially in the VIS and NIR regions. However, there is no mention about those at wavelength > 3 um. Authors should describe the spectral features in these regions as well.

*Thank you for pointing out this oversight. We've added a brief explanation of the generally constant low albedo and the minimal contribution to BBA due to the limited insolation at wavelengths > 3 μm.*

3. In general, the glacier surface tilted so there seems to be a limitation for the application of the plane-parallel RTM. It is desirable to mention the effect of slope on the spectral albedo calculations with the scope of application of SNICAR-ADv4.

*Thank you for pointing this out. In situations where the surface is slightly angled but still relatively smooth, plane-parallel approximations can be applied with an effective solar zenith angle adjusted for the slope and aspect of the surface. Other studies have analyzed the sensitivity of albedo to sloped and rough surfaces and developed methods to account for surface roughness and slope (Picard et al., 2020; Larue et al., 2020). However, these effects are out of the scope of the paper. A brief discussion on the limitations of the plane-parallel approximation has been added to the manuscript.*

4. Regarding the model evaluation against measured spectral albedo (Figs. 8-10),

- authors mentioned in the manuscript that for most of the comparisons, the exact conditions are unknown (L361). Thus, some of input data were determined (constrained) from the comparison between model and measurements. However, it is not clear what parameter(s) was(were) constrained in the RTM calculations. Authors should described these parameters in the manuscript to distinguish known parameters from the input data clearly.

*The model input parameters that were well constrained now have an asterisk next to them in Appendix A. We've also included more detailed descriptions in each model evaluation section (3.3.1-3.3.4) that indicate which variables were well constrained and which were not.*

- It seems that input parameters including the constrained ones shown in Appendix provide good agreement between model and measurement (Figs. 8-10) . However, there are no sufficient explanations in the manuscript as to how reasonable the constrained values are. Authors should give a careful explanation here.

*For each model – measurement comparison we utilized all the known model parameters from observations. All tuned or "unconstrained" model input variables were carefully chosen to both achieve good agreement between the model and measurements and also to be physically realistic. For unconstrained model parameters, we apply loose physical constraints that snow and ice follow with depth. For example, density and grain size increase with depth and LAC decreases with depth. We also utilize normal ranges of snow and ice properties, such as density and grain size/specific surface area, from measurements in similar regions. A more thorough explanation of how the unconstrained variables are chosen has been added to the revised manuscript.*

- In addition, spectral albedos were seemed to be calculated under fixed conditions as follows: the mid-latitude winter profile in the atmospheric profile (even in Greenland ice sheet), the solar zenith angle of 50 degree (for all cases) and the hexagonal plate as snow grain shape (even in large grain size). Reviewer knows that similar spectral albedos can be achieved using different model parameters (L322). But, I think these conditions do not suit the validation of the SNICAR-ADv4 even though authors showed good agreement between model and measurement. At least the latter two parameters should be determined (constrained) from the measurement values to calculate the spectral albedo properly.

*Thank you for pointing out this oversight. The SZA used in the model – measurement comparisons has been changed to be the SZA at solar noon for each measurement date and location. Because the measurements used do not include information about the shape of the snow grains, non-spherical grains are used in the modeling because spheres produce unrealistically large scattering asymmetry parameters. In this work, we default to the hexagonal plate shape as it has an intermediate asymmetry parameter between that of spheroids and Koch snowflake shaped grains (Flanner et al., 2021; He et al., 2017).*

Specific comments

L152: How did you marge two refractive indices in this analysis? You need explanations more in details.

*The merged ice refractive index is that described by Flanner et al. (2021). It utilizes the imaginary index of refraction from 0.2 – 0.6μm as reported by  Picard et al. (2016) Picard et al. (2016) and the real and imaginary index of refraction reported by Warren and Brandt (2008) elsewhere in the spectrum. For more clarity, we've removed "merged", referenced the description in Flanner et al. (2021), and included a brief explanation.*

L154: Definition of thin snow layer overlying ice is not clear. How thin is it? For example, give the optical thickness of thin snow layer.

*In this context, the model's ability to represent a thin snow layer, of any arbitrary thickness, overlying ice is being described. That has been made more clear in the revised text. Thank you for pointing this out. The impact of varying scattering layer depths and optical properties is described in more depth in section 3.1 surrounding the discussion of figure 5.*

L203: I don't know how this number oln(1.5) is valid for the size distribution of air bubble. Please give a valid explanation.

*We've added the Carras and Macklin (1975) citation which discusses previous studies finding that the size distribution of air bubbles within hailstones is lognormal. We also now reference the Dadic et al. (2013) ice bubble measurements, which show skewed bubble size distributions. Dadic et al., (2013) figure 10 shows that surface firn and ice favors smaller air bubbles (radius < 0.4 mm) and deep ice favors larger air bubbles (radius > 0.7 mm). While we are not aware of measurements supporting a geometric standard deviation of 1.5, the value assumed for the lognormal width is not particularly important. This is because the optical properties of air bubble distributions with identical specific surface area (or effective radius) are nearly identical, and we use effective radius as the descriptive variable for bubble size. The distribution just needs to be sufficiently large enough to average over Mie resonance features, and 1.5 is indeed large enough to achieve this. A brief explanation of the use of a lognormal size distribution with a standard deviation of 1.5 has been added to the revied manuscript.*

L313: "SNICAR-ADv4 simulates a wide … of snow and ice". It seems that this sentence has been taken out of context.

*We've adjusted the wording of this sentence to better relate it to the previous sentence.*

Figure 1: I can't see the shading range of spectral albedo especially for NIR regions because y-axis is so shrunk compared to x-axis. I recommend to replot (reshape) all figures (a-f) such as Fig. 3 or Fig. 6 for example.

*Thank you for pointing this out. We have reshaped the figures so they have a closer 1-1 x-y ratio.*

Figures 8, 9 and 10: I recommend to replot (reshape) all figures to see the spectral features clearly such as Fig. 3 or Fig. 6 for example. In addition, please show the difference between model and measurements in order to see how differences there are .

*Thank you for pointing this out. The figures have been reformatted so they have a closer 1-1 x-y ratio. We have also added the difference between the measurements and model albedo to each comparison plot. The difference is the modeled albedo value minus the measurement value interpolated to the higher resolution model $\lambda$ scale. Negative values indicate the model is underestimating the albedo and positive values indicate the model is overestimating the albedo. Figure 2 in the supplement is an example of the model - measurement comparison reformatted with the difference plot included.*

[Figure]

Citations:

Carras, J. N. and Macklin, W. C.: Air bubbles in accreted ice, Q.J Royal Met. Soc., 101, 127–146, https://doi.org/10.1002/qj.49710142711, 1975.

Dadic, R., Mullen, P. C., Schneebeli, M., Brandt, R. E., and Warren, S. G.: Effects of bubbles, cracks, and volcanic tephra on the spectral albedo of bare ice near the Transantarctic Mountains: Implications for sea glaciers on Snowball Earth, JGR: Earth Surface, 118, 1658–1676, https://doi.org/10.1002/jgrf.20098, 2013.

Flanner, M. G., Arnheim, J., Cook, J. M., Dang, C., He, C., Huang, X., Singh, D., Skiles, S. M., Whicker, C. A., and Zender, C. S.: SNICAR-AD v3: A Community Tool for Modeling Spectral Snow Albedo, GMD, 1–49, https://doi.org/10.5194/gmd-2021-182, 2021.

He, C., Takano, Y., Liou, K.-N., Yang, P., Li, Q., and Chen, F.: Impact of Snow Grain Shape and Black Carbon–Snow Internal Mixing on Snow Optical Properties: Parameterizations for Climate Models, J. Climate, 30, 10019–10036, https://doi.org/10.1175/JCLI-D-17-0300.1, 2017.

Larue, F., Picard, G., Arnaud, L., Ollivier, I., Delcourt, C., Lamare, M., Tuzet, F., Revuelto, J., and Dumont, M.: Snow albedo sensitivity to macroscopic surface roughness using a new ray-tracing model, TC 14, 1651–1672, https://doi.org/10.5194/tc-14-1651-2020, 2020.

Picard, G., Libois, Q., Arnaud, L., Verin, G., and Dumont, M.: Development and calibration of an automatic spectral albedometer to estimate near-surface snow SSA time series, TC, 10, 1297–1316, https://doi.org/10.5194/tc-10-1297-2016, 2016.

Picard, G., Dumont, M., Lamare, M., Tuzet, F., Larue, F., Pirazzini, R., and Arnaud, L.: Spectral albedo measurements over snow-covered slopes: theory and slope effect corrections, TC, 14, 1497–1517, https://doi.org/10.5194/tc-14-1497-2020, 2020.

Warren, S. G. and Brandt, R. E.: Optical constants of ice from the ultraviolet to the microwave: A revised compilation, JGR, 113, https://doi.org/10.1029/2007JD009744, 2008.

*Dr. Dadic, thank you for your review of our paper and your insightful comments. All comments are addressed below, with the original comment first and the response in italics.*

Review #2 (Ruzica Dadic):

Summary and recommendations

This paper uses extends the current multi-layer two-stream delta-Eddington radiative transfer model to be applicable in a wide range of snow and ice environments. The work is relevant because of the role that cryospheric albedo plays in Earth's climate in the context of climate change. Especially the inclusion of LACs is of relevance and I appreciate that the authors include different types of impurities, so the model really is applicable in a wide range of regions. The paper is well written with and the methods are well explained. I only have a few minor comments, that I summarize below.

1) There are inconsistencies on the definition of "snow" and "ice" in the model. In the abstract, the cutoff is 650 kg/m^3. In Figure 2, density of 600 is defined as ice. And later you say that snow is below 500 and ice is above 600? How is the firn modeled, as firn or as ice? This just needs to be made consistent. I think it's just adjustments in the text, the results appear to be ok.

*Thank you for pointing out this inconsistency. We originally used 600 kg/m3 to show a potential range of ice albedo. This has been addressed by changing the density threshold to be 650 rather than 600 for figures 2, 3, 6 and 7. Firn can be represented as either high density snow with a large snow grain size or low density ice with a small air bubble radii. This is discussed in the manuscript:*

*L316: "Firn has an intermediate density and can be treated as snow or ice, allowing for the techniques to be compared for media with equivalent SSAs. From a modeling perspective, it would be useful to specify a density threshold for representing a layer as snow or ice, as the model is sensitive to the ice density, and density is more easily measured in the field than other physical properties. Because ice is represented as air bubbles within snow it could be valid to treat all firn with a density greater than half that of pure ice (458.5 kg $m^{-3}$) as an ice layer. However, it is unlikely that ice that porous necessitates a refractive boundary. The transition between firn to ice is where pores between ice grains close and form air bubbles within a solid ice media. The closing-off of air bubbles occurs at an ice density around ~830 kg $m^{-3}$ or when ~10% of the ice volume is composed of air bubbles (Bender et al., 1997; Dadic et al., 2013). Because SNICAR-ADv4 incorporates numerous parameters, such as the density, grain size, layer depth, and the inclusion of an SSL, similar spectral albedos can be achieved using different model parameters (further described in Section 3.3.2). We see greater agreement between model and measurement for layers represented as ice with densities between 650-700 kg $m^{-3}$ (Fig. 10b) and recommend that users treat media with densities over 650 kg $m^{-3}$ as ice layers."*

*In figure 10b, when comparing SNICAR-ADv4 to the Dadic et al. (2013) firn measurements, we found that using an ice layer rather than a snow layer achieved slightly better agreement, especially at $\lambda > 1.2$ μm.*

2) Usually the term SSL (surface scattering layer) is used just for sea ice and shows a particular structure because of the anisotropic structure of brine channels in sea ice. To avoid confusion, you might considering calling it a different name, because the "crusty layer" on glacier ice is not associated with brine channels and is more sotropic that the same layer on sea ice.

*Thank you for pointing out the specificity of the term SSL as applied in the sea ice community. To avoid confusion with this context of SSL, we have decided to instead refer to this layer as a "rough scattering layer". The main purposes of the layer are to 1) add surface roughness that reduces or eliminates specular reflection by the Fresnel Layer and 2) introduce small scale surface roughness typical of snow and ice surfaces. A glacial crustal surface, which is a very coarse and porous ice surface, can be roughly represented using very low density snow and large aspherical snow grains. SNICAR-ADv4 also includes the ability to simulate scattering layers that are optically dissimilar from glacial crustal surfaces, therefore, we avoid calling it a "crustal layer" and use "rough scattering layer" as a more general term.*

3) You say that snow layers are represented as "ice crystals", but it's rather "spheres". I would correct this throughout the manuscript, unless you are really representing the crystal shape instead of a collection of independent spheres.

*The text has been changed from "crystals" to "grains" because this study utilizes hexagonal plate shaped ice grains. The model allows for spheroids, hexagonal plates, or Koch snowflake shaped grains. We've included a more thorough description of the use of hexagonal plates in the manuscript in response to R1 comment 4.3.*

4) L 153: what is the "merged" ice refractive index?

*The merged ice refractive index is described in Flanner et al., (2021). It utilizes the imaginary index of refraction from 0.2 – 0.6µm as reported in Picard et al. (2016) and the real and imaginary index of refraction reported by Warren and Brandt (2008) elsewhere in the spectrum. For more clarity, the word "merged" has been removed, Flanner et al. (2021) is referenced in the description, and a brief explanation is included.*

5) Figures 8-11 are hard to read, because the x-axis is so stretched. Maybe redo them with a better readable x-y-ratio. Also it may be worth plotting the difference (in %) between model and measurements at different wavelengths, rather than actual values. Or add the differences as a righthand y-axis. It may be worth to give the %-ages of albedo variations in the manuscript, rather than absolute values (e.g. L24, L288, L291)

*(Same response to R1 – last comment) Thank you for pointing this out. The figures have been reformatted so they have a closer 1-1 x-y ratio. We have also added the difference between the measurements and model albedo to each comparison plot. The difference is the modeled albedo value minus the measurement value interpolated to the higher resolution model λ scale. Negative values indicate the model is underestimating the albedo and positive values indicate the model is overestimating the albedo. Figure 2 in the supplement is an example of the model - measurement comparison reformatted with the difference plot included.*

*We chose to show the absolute difference to avoid inflating the difference at longer wavelengths where the albedo is generally lower due to higher absorption and limited insolation.*

*Please see the Figure 2 in the supplement for R1 comment's for the new measurement and model comparison plots.*

*Citations:*

*Flanner, M. G., Arnheim, J., Cook, J. M., Dang, C., He, C., Huang, X., Singh, D., Skiles, S. M., Whicker, C. A., and Zender, C. S.: SNICAR-AD v3: A Community Tool for Modeling Spectral Snow Albedo, GMD, 1–49, https://doi.org/10.5194/gmd-2021-182, 2021.*

*Picard, G., Libois, Q., Arnaud, L., Verin, G., and Dumont, M.: Development and calibration of an automatic spectral albedometer to estimate near-surface snow SSA time series, The Cryosphere, 10, 1297–1316, https://doi.org/10.5194/tc-10-1297-2016, 2016.*

*Warren, S. G. and Brandt, R. E.: Optical constants of ice from the ultraviolet to the microwave: A revised compilation, JGR, 113, https://doi.org/10.1029/2007JD009744, 2008.*

---

## Author Response (AR2)

Dear Dr. Mölg,

Thank you for overseeing and accepting the publication of this paper.

Thank you for catching the incomplete figure caption. We have added detail to the caption, included more detail in Table A1, and fixed the legend in figure 2.

Best,
Chloe Whicker and coauthors